

# Field and laboratory evaluation of a high time resolution x-ray fluorescence instrument for determining the elemental composition of ambient aerosols

5   Anja H. Tremper[1], Anna Font[1], Max Priestman[1], Samera H. Hamad[2], Tsai-Chia Chung[3], Ari Pribadi[1], Richard J. C. Brown[4], Sharon L. Goddard[4], Nathalie Grassineau[3], Krag Petterson[5] Frank J. Kelly[1] and David C. Green[1]

[1] MRC-PHE Centre for Environment and Health, King's College London, London, SE1 9NH, United Kingdom
[2] Department of Behavioural and Community Health, School of Public Health, the University of Maryland,
10  College Park, MD 20742, USA
[3] Earth Sciences Department, Royal Holloway University of London, Egham TW20 0EX, UK
[4] Chemical, Medical and Environmental Science Department, National Physical Laboratory, Teddington, TW11 0LW, UK
[5] Cooper Environmental Services, LLC, 9403 SW Nimbus Ave. Beaverton, OR 97062

15  *Correspondence to*: Anja H. Tremper (anja.tremper@kcl.ac.uk)



**Abstract.** Measuring the chemical composition of airborne particulate matter (PM) can provide valuable information on the concentration of regulated toxic metals and their sources and assist in the identification and validation of abatement techniques. Undertaking these at a high time resolution (1 hour or less) enables receptor modelling techniques to be more robustly linked to emission processes. This study describes a comprehensive

laboratory and field evaluation of a high time resolution x-ray fluorescence (XRF) instrument (CES XACT 625) for a range of elements (As, Ba, Ca, Cd, Ce, Cl, Cr, Cu, Fe, K, Mn, Mo, Ni, Pb, Pd, Pt, S, Sb, Se, Si, Sr, Ti, V, Zn) against alternative techniques: high time resolution mass measurements, high time resolution ion chromatography, aerosol mass spectrometry, and established filter-based, laboratory analysis using inductively coupled plasma mass spectrometry (ICP-MS). 1) Laboratory evaluation was carried out using a novel mass-

based calibration technique to independently assess the accuracy of the XRF against laboratory generated aerosols, which resulted in slopes that were not significantly different from unity. This demonstrated that generated particles can serve as an alternative calibration method for this instrument. 2) The XACT was evaluated in three contrasting field deployments; a heavily trafficked roadside site ($PM_{10}$ and $PM_{2.5}$), an industrial location downwind of a nickel refinery ($PM_{10}$) and an urban background location influenced by

nearby industries and motorways ($PM_{10}$). The XRF technique agreed well with the ICP-MS measurements of daily filter samples in all cases with a median $R^2$ of 0.93 and a median slope of 1.07. Differences were likely due to recovery rates from the sample digestion as well as filter sampling artefacts and matrix effects in the XRF technique. The XRF technique also agreed well with the other high time resolution measurements but showed a significant positive bias (slopes between 1.41 and 4.6), probably due to differences in the size selection

methodology, volatility and water solubility of the PM in aerosol mass spectrometry and ion chromatography, respectively. 3) A novel filter analysis technique using the XACT showed promising initial results: filters analysed off-line with the XACT compared well to in-situ XACT measurements with a median $R^2$ of 0.96 and median slope of 1.07. The resulting range of slopes was comparable to slopes produced in the ICP-MS comparison. This technique provides an opportunity to use the XACT when it is not deployed in the field; thus

expanding the potential use of this instrument in future studies.

## 1    Introduction

It has long been known that increased air pollution, and specifically particle pollution is associated with adverse health effects (Brunekreef and Holgate, 2002; Kelly et al., 2012). Particulate matter (PM) also affects atmospheric visibility and radiative forcing (Fuzzi et al., 2015). PM is not a homogenous air pollutant but rather

a complex mixture; it varies in chemical and physical composition depending on the contributing sources and the atmospheric processes (WHO, 2000). The composition of PM influences its harmfulness and therefore it is important to gain better knowledge about which chemical components might cause particle toxicity (Kelly and Fussell, 2015). Understanding the chemical composition of PM also provides information on the sources and thus helps implement policies on targeting these emission sources (WHO, 2013). Trace metals in particular,

even though they do not contribute substantially to the mass of PM, act as markers for specific source categories (Visser et al., 2015a) and evidence is emerging that some metals in ambient PM are associated with adverse health effects at concentrations near to current ambient levels (Chen and Lippmann, 2009).

Accurate measurements of the PM composition are important and are mostly carried out by collecting PM on filters using high or low volume filter samplers (e.g. Digitel-DAH-80, Partisol 2025) and subsequently digesting





and analysing these in a laboratory. These filters are collected over a period of time, usually 24 hours to a week, and then analysed for different components such as metals (Brown et al., 2008), polyaromatic hydrocarbons (Pandey et al., 2011), elemental and organic carbon (Chu, 2004). This approach is time consuming, labour intensive and prone to positive and negative sampling artefacts (Chow et al., 2015). Also, it only gives

compositional information with a considerable time delay and at low temporal resolution which cannot be effectively associated with meteorological variability or short term variations in emissions.

To run the above filter samplers on a higher time resolution means they become even more labour intensive to operate. To address this limitation, sampling devices were developed to collect PM either hourly or sub-hourly without the need for frequent filter changes. These include the Rotating Drum Impactor (Bukowiecki et al.,

2005) which collects three size ranges: $PM_{10-2.5}$ (coarse), $PM_{2.5-1.0}$ (intermediate) and $PM_{1.0-0.3}$ (fine), by passing sequentially through three rectangular nozzles of decreasing size; and the Streaker (PIXE International Corporation) which consists of two collecting substrates rotating at constant speed producing a circular continuous deposition of both $PM_{10-2.5}$ and $PM_{2.5}$ (Formenti et al., 1996). Nevertheless the analysis is still performed in the laboratory and thus does not improve the time delay of the analysis.

Several online high time resolution instruments have also been developed in recent years which address some of the sampling artefact, resource and time resolution limitations of laboratory approaches. These include aerosol mass spectrometers such as the ACSM (Aerodyne Research Inc.) (Ng et al., 2011); ion chromatography approaches such as the MARGA (Metrohm) (Rumsey et al., 2014), PILS (Brechtel) (Weber et al., 2001) and URG's 9000 ambient ion monitor (Beccaceci et al., 2015); and x-ray fluorescence (XRF) such as the XACT

instrument (Cooper Environmental Services) (Park et al., 2014). However, these high time resolution instruments only measure a subset of chemical components each, depending on their collection, extraction and analysis methodology. Therefore multiple collocated instruments are needed to measure the full PM composition. Furthermore, the high time resolution instruments tend to sample a narrower range of components with a higher Limit of Detection (LOD) than equivalent laboratory based methods, generally because less

material is collected on each sample. For example, the synchrotron radiation-induced XRF (SR-XRF) used by Visser et al. (2015b) measured elements with atomic numbers greater than 11 while the XACT measures elements with atomic numbers greater than 14 thereby missing important contributors to PM mass such as Na, Mg and Al; the LODs reported for the SR-XRF analysis (Visser et al., 2015b) are generally lower than those for the XACT (Furger et al., 2017; Park et al., 2014).

Despite these limitations, the XACT is unique in measuring elements automatically using energy dispersive XRF (ED-XRF) and has been successfully evaluated in a number of field studies (Furger et al., 2017; Park et al., 2014). Park et al. (2014) found a good agreement between the XACT and 24 hour filter based measurement collected in South Korea (filters analysed using ED-XRF). Furger et al. (2017) tested the XACT during a summer campaign in Switzerland in 2015 and compared the XACT data with measurements made using ICP-

MS on filters sampled for 24 hours (both $PM_{10}$) as well as ACSM measurements ($PM_1$). They found an excellent correlation, with $R^2$ values $\geq 0.95$, between the XACT and ICP-MS data for ten elements (S, K, Ca, Ti, Mn, Fe, Cu, Zn, Ba, Pb). However, they found that the XACT was systematically higher than the filter based technique. In Jeong et al. (2017) hourly trace elements measured by the XACT were included in positive matrix factorisation (PMF), which allowed a more robust apportionment of PM sources (Jeong et al., 2017).



For all analytical techniques, in the field and laboratory, the confidence in measurements largely depends on high quality, traceable calibration of the instruments (Indresand et al., 2013). In the case of the XACT, the calibration is carried out using thin film standards, which are thin element films deposited on Nuclepore substrates and are available for elements between atomic number 11 and 82 (EPA Compendium Method IO-3.3 for the Determination of Inorganic Compounds in Ambient Air, EPA/625/R-96/010a, Table 2, page 3.3-16). This is an established method but has been reported to have various limitations (Indresand et al., 2013): the standards are much higher in concentration than most ambient samples; the element mix of the standard might not be representative of ambient particle mix; and the collection properties on a filter may also differ. Alternative calibration methods have therefore been tested to address these issues. For example Indresand et al., (2013) produced sulphur reference materials that replicated PM samples to successfully calibrate XRF systems.

In this study a novel mass-based calibration technique for the XACT 625 has been developed to independently assess the accuracy of the XRF method for a range of elements at more atmospherically relevant concentrations. This study also reports the field evaluation of the XACT at both traffic and industrial sites in the UK where it was compared to independent measurements of $PM_{2.5}$ and $PM_{10}$ on daily filters, analysed by ICP-MS; and also to alternative high time resolution chemical speciation instruments (ion chromatography and aerosol mass spectrometry). Additionally, the ability of the XACT to analyse $PM_{10}$ filter samples in the laboratory was piloted and the results compared to collocated in-situ XACT measurements. Using the instrument in this way potentially diversifies experimental sampling programmes with this single resource by deploying additional sampling devices.

## 2 Materials and Methods

### 2.1 XACT 625

The instrument measures 24 elements between Silicon and Uranium at a time resolution between 15 minutes and four hours using ED-XRF. The size fraction of the PM sample collected onto the Teflon filter tape depends on the size selective inlet chosen. The instrument samples with a volumetric flow rate of 1 $m^3$ $h^{-1}$ through an inlet tube heated to 45 ˚C when the ambient relative humidity (RH) exceeds 45% to avoid water depositing on the tape. Sampling and analysis is performed continuously and simultaneously, except for the time required to advance the filter tape (~20 s) from the sample to the analysis position. Daily automated quality assurance checks are performed every night at midnight and consist of an energy alignment (an energy calibration using a copper rod, inserted into the analysis area); and upscale measurement to monitor the stability of the instrument response (for Cd, Cr and Pb); and a flow check through an independent mass flow sensor. Additional quality assurance checks employed here included flow calibrations, regular external standard checks, field blanks performed using a HEPA filter as well as tape blanks before and after each tape change.

For the field studies the instrument sampled $PM_{10}$ or $PM_{2.5}$ as detailed below (see Sect. 2.3.1). The elements reported are As, Ba, Ca, Cd, Ce, Cl, Cr, Cu, Fe, K, Mn, Mo, Ni, Pb, Pt, S, Sb, Se, Si, Sr, Ti, V, Zn and were chosen to represent a range of source categories (i.e. regulatory, traffic, industry), plus the internal Palladium (Pd) standard. The internal standard measurement is the reported response from a Pd rod inserted in a fixed position under the filter tape.



## 2.2 Laboratory Experiments

An independent mass-based calibration technique was developed for the XACT. This used laboratory generated aerosols and a schematic of the instrument set-up is shown in Figure 1. Ammonium sulphate ($NH_3SO_4$, ACS reagent grade, Sigma-Aldrich), potassium chloride (KCl, analytical grade, VWR Chemicals) and zinc acetate

($Zn(O_2CCH_3)_2$, analytical grade, VWR Chemicals) were dissolved in high purity water (18.2 MΩ, TOC < 5ppb, PURELAB® Ultra Analytic, ELGA (Veolia Water Technologies)) to obtain a range of standard solutions spanning the ambient concentration range.

Aerosols were generated using an ATM 226 - Clean Room Aerosol Generator (Topas) and were driven through two Permapure driers set in reflux method to reduce the relative humidity to approximately 40%. The flow was

then split isokinetically using a TSI 3708 flow splitter and passed to three instruments: a tapered element oscillating microbalance (1400ab TEOM, Thermo), with which continuous direct mass measurements of particulates were taken; a scanning mobility particle sizer (TSI SMPS 3080); and the XACT. HEPA filtered make-up air was provided where necessary. The mass concentration of the deposited $(NH_4)_2SO_4$, KCl and $Zn(O_2CCH_3)_2$ as measured by the TEOM were used to calculate the S, Cl, K and Zn mass concentrations and

compared to the element concentration measured with the XACT. The SMPS was used to give qualitative diagnostic information on the size distribution of the aerosol.

## 2.3 Field Experiments

### 2.3.1 Monitoring Locations

Three field evaluation campaigns were carried out in the UK (Table 1): a traffic site in central London

(Marylebone Road: lat 51°31'21"N, long 0°09'17"W) and two industrial sites (Pontardawe in Wales: lat 51°43'12"N, long 3°50'49"W; and Tinsley in Sheffield: lat 53°24'38"N, long 1°23'46"W) (map in Supplement S1). Marylebone Road is a kerbside monitoring station in a central London street canyon adjacent to a six lane highway (60-80,000 vehicles per day). During this deployment the XACT sampled $PM_{10}$ except for a period from October to December 2014 that sampled $PM_{2.5}$. Pontardawe is an urban industrial site in South Wales,

surrounded by metallurgical industries. Tinsley, located north-east of Sheffield, is approximately 200 m east of the M1 motorway, with a residential area to the east and light industry to the west. In Pontardawe and Tinsley, the XACT was collocated with the monitoring site belonging to the UK Ambient Air Quality Metals Monitoring Network from which daily filters measured by ICP-MS were available.



### 2.3.2    Comparison Instruments

A number of comparison instruments were used to evaluate the XACT in the field. The main comparison was carried out using filter samples collected with a Partisol 2025 and subsequent ICP-MS analysis. Further, an Aerosol Chemical Speciation Monitor (ACSM) and Ambient Ion Monitor-URG-900B (URG) were used for the
evaluation of XACT at a high time resolution. Although the measurands are not directly comparable, they provide useful information for studies where source contributions may be assumed based on one of these measurement techniques.

**Partisol 2025**

A Thermo Scientific Partisol 2025, with a flow rate of 1 m³ h⁻¹, was used to collect filter samples (mixed
cellulose ester filters, VWR 514-0464) for subsequent analysis using ICP-MS. At Marylebone Road, where samples were taken specifically for this study, a 23 hour sampling period was used (01:00-00:00) to ensure comparability with the XACT once the equivalent hour lost to quality assurance was removed. The filters were acid-digested on a hotplate using a 1:2 mixture of $HClO_4$ and HF in open 10ml Teflon crucibles. After complete evaporation, $HNO_3$ has been added to each sample, and the remaining solution was made up to the required
volume. Filters were fully dissolved with this method (adapted from ISO-14869-1:2001). For quality assurance, blank filters (field and laboratory blanks), internal (rhyolite) and international (NIST SRM 1648a) certified reference materials were also prepared following the same procedure. The samples were analysed for a range of elements using ICP-MS (Table 3).

At Pontardawe and Tinsley, where an established measurement programme was adapted for comparison, a 24
hour period was sampled. Thus the frequency of $PM_{10}$ filter sampling at the adjacent UK Heavy Metals Network sites was increased from weekly to daily for these field evaluations. The filters were digested using $HNO_3/H_2O_2$ digestion following the European reference method EN14902 and analysed for a range of elements (Table 4 and Table 5) using ICP-MS (Goddard et al., 2016 ).

The certified reference material was used for quality control in both filter digestion protocols. As standard
reference materials are usually not an exact match for the matrix of the sample, the resulting recovery rates serve as a quality control parameter rather than a calibrant. Samples were thus not corrected for the recovery rate but checked for compliance with the requirements described in EN14902; recovery rates for both digestions methods are given in S5.

**Aerosol Chemical Speciation Monitor (ACSM)**

The ACSM measured the chemical composition of non-refractory $PM_1$ ($NO_3$, $SO_4$, $NH_4$ and organic mass) and is fully described in Ng et al. (2011). Briefly, air was drawn through an URG $PM_{2.5}$ size selective inlet (URG-2000-30EQ) at 0.18 m³h⁻¹ and subsequently dried using a Permapure™ drier (Perma Pure PD Dryer, PD-07018T-12MSS). Particles were focused using an aerodynamic lens with a 50% transmission range of 75–650 nm (Liu et al., 2007) and subsequently flash vaporised, ionised and analysed using mass spectrometry at 0-100
amu. The signal was resolved into $NO_3$, $SO_4$, $NH_4$ and organic mass using a library of known fragmentation characteristics. The aerosol was sampled and analysed alternately with background air, allowing a continuous air subtraction, and averaged to an hourly time resolution. The ionisation efficiency was calculated using a mono-disperse supply of ammonium nitrate aerosols that were size selected through a differential mobility analyser and counted using a condensation particle counter (CPC). The relative ionisation efficiencies of
sulphate and ammonium were calculated from separate calibrations using a mono-disperse supply of ammonium



sulphate aerosols. The collection efficiency was calculated using the Middlebook parameterisation
(Middlebrook et al., 2012), which calculates an optimum collection based on aerosol acidity, inlet humidity and
particle composition. The measurements were quality assured against measurements of SMPS (for volume to
ensure the collection efficiency is suitable) and $PM_{2.5}$ mass when combined with Aethalometer measurements as
described by Crenn et al. (2015).

**Ambient Ion Monitor- URG-900B (URG)**

The URG-900B Ambient Ion Monitor continuously measured water-soluble anion and cation concentrations
($Cl^-$, $SO_4^{2-}$, $NO_3^-$, $Na^+$, $NH_4^+$, $K^+$, $Mg^{2+}$, and $Ca^{2+}$) in $PM_{10}$ and is described in (Beccaceci et al., 2015). Briefly,
the sample was drawn at a flow rate of 1 $m^3\,h^{-1}$ through a size selective inlet (PM10); the sample was then split
isokinetically through a flow splitter to allow a 0.18 $m^3\,h^{-1}$ flow into a liquid diffusion denuder containing $H_2O_2$
to remove interfering acidic and basic gases. The remaining particles in this air stream were then enlarged in a
super saturation chamber and finally collected in an aerosol sample collector and injected into the (anion and
cation) ion chromatographs every hour.

### 2.4   Laboratory based filter analysis using the XACT

To trial a filter analysis technique using the XACT, $PM_{10}$ was sampled onto polytetrafluoroethylene (PTFE)
filters (Zefluor, 0.5μm, 47mm disc, Pall Life Sciences 516-8908) for 24 hours using a Partisol 2025 during the
field campaign in Sheffield in February/March 2017. These PTFE filters were a similar material to the XACT
filter tape but the stronger structure enables easier handling during punching and analysis. After exposure a
25mm punch was taken out of the exposed filters for analysis with the XACT on its return to the laboratory. The
punching tool was always aligned with the edge of the exposed area. The punch was transferred into a filter
holder, identical to the one used for instrument calibration with thin film standards, and transferred into the
holder slot in the analysis block of the XACT. The analysis was performed on a 15 minute sample time using
the XRF control program in a manual analysis mode. The energy condition set up remained the same as during
the field sampling in the automation mode. Each filter was analysed four times, and the filter punch was rotated
90 ° in the filter holder in between replicates in order to account for non-uniformity of the particle deposit on the
filter punch. The XACT results were used to calculate daily ambient element concentrations, which were
compared to the daily mean concentration measured by the XACT in-situ. A total of 12 filters were analysed.

### 2.5   Regression Analysis Approach

All comparisons were carried out using the Deming regression which minimises the sum of distances between
the regression line and the *X* and *Y* variables taking into account the uncertainties in both variables (Deming,
1943).

### 2.6   Treatment of Measurements below Limit of Detection

In all comparisons data under the detection limit was used as measured unless the value was zero or below, in
which case 0.5*LOD was used to replace the value. Including values below the LOD had the advantage of being
able to include daily XACT mean concentration was calculated from hourly concentrations that might have been
lost if data below LOD was excluded and the daily data capture was not met.





### 2.7 Uncertainty Evaluation

The expanded uncertainty, representing a 95 % level of confidence, was calculated by taking the root of sum square of the separate sources of uncertainty as shown below:

$$U = \sqrt{LOD_i{}^2 + (b.c_i)^2}$$

Where $LOD_i$ is the limit of detection of element $i$ (here calculated as the 3 times the experimental standard
deviation of field or laboratory blanks), $c_i$ is the measured concentration of the element (in ng m$^{-3}$), and $b$ is an element dependent factor which was derived from experimental and literature values (US-EPA, 1999). For the XACT measurements, the combined uncertainty included contributions of 3/√3% from flow (CEN, 2014), 5% from calibration standard uncertainty (US EPA, 1999), 2.9% from long term stability (calculated from the standard deviation of hourly internal Pd reference) and an element-specific uncertainty associated with the
spectral deconvolution calculated by the instrument software for each spectra. The XACT LOD was determined using HEPA field blank measurements during each campaign; these are shown in Table 3. . For the ACSM, the sulphate measurement uncertainty was estimated as 14% (k=2) for sulphate at a 30 min resolution by Crenn et al. (2015) and the LOD was determined using HEPA field blank measurements as 34.9 ng m$^{-3}$.For the URG, the chloride and sulphate LODs were reported by the manufacturer as 100 ng m$^{-3}$ and verified by Beccaceci et al.
(2015). The uncertainty of the species measured by ion chromatography was estimated at 4.5% (k=2) by Yardley et al. (2007) and combined with the additional 97% extraction efficiency of a particle-to-liquid sampler system estimated by Orsini et al. (2003).

### 3 Results and Discussion

### 3.1 Laboratory Experiment

For the calibration test a range of solution concentrations were produced to assess the instrument response (see supplement S2). A subset of concentrations, which span the concentrations encountered during the field campaign, was used for the final comparison (see Table 2). Thus, the highest element concentrations in the standards used for comparison were between 9 (S) and 25 (Zn) times lower than the commercial thin film standards.

All calibrations resulted in a linear relationship between the mass calculated using TEOM mass concentrations and measured by the XACT for the standard range used. TEOM and XACT results agreed well in all cases with slopes between 0.94 and 0.99. Slopes are not significantly different from the 1:1 line for all comparisons. The coefficient of determination (R$^2$) ranged between 0.98 (S) and 0.99 (Cl, K, Zn). The XACT response to the generated particles was thus comparable to the response of the commercial standards used for calibration. A
similar result was found by Indresand et al. (2013) using prepared sulphur reference materials for XRF calibration.

### 3.2 Field Evaluation: overview

An overview of the data recorded in each comparison is given in Table 3-6 and includes the limit of detection for all elements. Sb was not included in the analysis as spectral interference resulted in a high LOD.



The sampling at Marylebone Road was carried out using a $PM_{2.5}$ inlet during a period that was dominated by fireworks activity (Oct-Dec 2014). The mean concentrations across all elements measured during this campaign ranged from 0.177 to 600 ng m$^{-3}$ and elements typically used in fireworks such as Ba, Sr, K and Ti had high maximum concentrations. Traffic emissions further influenced the metal concentrations at Marylebone Road.

Overall the order of the elements in terms of mean concentration was:

S > Fe > Cl > K > Si > Ca > Zn > Cu > Ba > Pb > Mn > Ti > Cd > Sr > As > Cr > Ce > V > Ni > Mo > Pt > Se.
This dataset helps highlight that high time resolution data has the advantage of giving much more detailed information on high pollution events, which can be used e.g. for health studies (Hamad et al., 2016). Figure 3 shows the daily filter and hourly XACT measurements of K and Ba during a period of increased bonfire and

fireworks activity due to Diwali (Hindu festival of light) and Guy Fawkes celebrations. The daily filter measurements show that the highest concentrations of K, which is used as an oxidiser in fireworks but also a tracer for biomass burning, were measured on the 5$^{th}$ and 6$^{th}$ November 2014, followed by slightly lower concentrations on the 7$^{th}$ and 8$^{th}$ of November. On the other hand Ba, which is used in green fireworks, displays similarly high concentrations on all four days. Looking at the K concentration in a higher time resolution as

measured by the XACT, it is evident that peak concentrations were comparable on the nights of the 5$^{th}$, 7$^{th}$ and 8$^{th}$ of November (data is missing for the 6$^{th}$ of November due to instrument failure) but the high concentrations did not last as long on the 7$^{th}$ and 8$^{th}$ of November. The highest Ba concentration on the other hand was measured on the 8$^{th}$ of November with lower concentrations on the 5$^{th}$ and 7$^{th}$. This difference in contribution might point to different fireworks being used.

Sampling at Pontardawe, Wales was carried out in an area dominated by metallurgical industry, which is reflected by the high Nickel concentrations measured (i.e. the mean Nickel concentration at Pontardawe was 27 times higher than that measured at Marylebone Road). The mean concentrations measured in this campaign ranged from 0.24 to 5,200 ng m$^{-3}$. The concentrations and dominant elements will be influenced by the site characteristics as well as the size range sampled; e.g. Cl from sea salt is predominantly found in the coarse

fraction and thus much higher at Pontardawe as sampling was carried out using a $PM_{10}$ head. The order of elements in terms of mean concentration in Wales was:

Cl > S > Si > Fe > Ca > K > Ni > Ti > Zn > Cu > Pb > Mn > Cd > Sr > Cr > Ba > Mo > V > Ce > As > Pt > Se.
In Wales, the availability of high time resolution data, in conjunction with meteorological data and source emission activity allowed us to pinpoint pollution sources more accurately. Cr concentrations from local sources

were studied to identify contributions from different industries. As can be seen in Figure 4 the 24 hour filter data leads to very different source directions than the higher time resolution data by the XACT (Font et al., 2017). This could be used to address policy breaches with more targeted abatement measures.

The influence of the local industry in Tinsley, Sheffield was reflected by high concentrations of metals like Ni and Cr, with mean concentrations more than 30 times that found in the Marylebone Road campaign with mean

concentrations ranged from 0.186 to 1,370 ng m$^{-3}$. The order of elements in terms of mean concentration in Tinsley was:

Cl > S > Fe > Ca > Si > K > Zn > Cr > Mn > Ni > Ti > Pb > Cu > Mo > Cd > As > Ba > V > Sr > Se > Ce > Pt.



The mean concentration of non-sea salt sulphate (XACT) and non-refractory sulphate (ACSM) during the fireworks campaign at Marylebone Road was 2,600 and 2,000 ng m$^{-3}$, respectively, with concentration ranging from 240 to 10,500 ng m$^{-3}$ SO$_4$ (non-sea salt) and 58 to 8,300 ng m$^{-3}$ for non-refractory SO$_4$.

The comparison of the XACT with the URG was carried out in PM$_{10}$ during winter 2014/2015. The concentration of water soluble anions and cation ranged from 154 (K) to 1,790 ng m$^{-3}$ (Cl) compared to 145 (K) to 2,700 ng m$^{-3}$ (Cl) in total element concentrations.

### 3.2.1    Comparison with ICP-MS

The filter comparison results were split by the two ICP-MS digestion methods: HF/HClO$_4$ and HNO$_3$/H$_2$O$_2$. This had the additional advantage of grouping the two industrial campaigns that were carried out in PM$_{10}$ and separating the campaign at Marylebone Road in PM$_{2.5}$. LODs were not consistently higher for either the ICP-MS or the XACT measurements (Table 3 -5). All elements were compared using Deming regression and a summary of all calculated slopes and intercepts are given in Table 7 (including R$^2$ values); the corresponding figures are available in the supplementary information section. The XACT agreed well with the ICP-MS measurements and R$^2$ ranged from 0.50-1.00 and 0.67-0.99, with a median of 0.91 and 0.95, following HF/HClO$_4$ and HNO$_3$/H$_2$O$_2$ digestion, respectively. Deming regression for Fe resulted in slopes that were not significantly different from unity for either subset. Slopes were also not significantly different from unity for Ba, Ca, K, Mn and Ti following digestion with HF/HClO$_4$ and for Cr, Ni, Pb, V and Zn following digestion with HNO$_3$/H$_2$O$_2$. For the element As the XACT recorded significantly higher concentrations than those measured by ICP-MS, irrespective of digestion method. This was also the case for elements Cu, Pb, Sr and Zn after HF/HClO$_4$ digestion and for Mn after HNO$_3$/H$_2$O$_2$ digestion. For the remaining elements (Ni after HF/HClO$_4$ digestion and Cu and Se after HNO$_3$/H$_2$O$_2$ digestion) the concentrations measured by the XACT were significantly lower than those measured by the ICP-MS. Cr and V were not reported for HF/HClO$_4$ due to contamination of the HClO$_4$ used in the digestion. The remaining elements were mostly below the limit of detection and thus did not produce meaningful regression results.

There are a variety of possible reasons for the differences observed between the methods: In the case of the filter analysis, the blank filters were found to be variable and thus subtracted values may result in an under- or overestimation of the true concentration; the digestion recovery rates were not taken into account; positive and negative filter artefacts could also influence the concentrations when sampling onto filters; many concentrations were close to the detection limit for the elements As in both campaigns and Ni during the Marylebone Road campaign. These stated reasons might influence the two digestions methods to different extends. Unfortunately, there was no opportunity to undertake both digestions on the same samples. To provide some insight into how the two digestion methods compared, the XACT measurements were grouped into concentration appropriate bins and the associated ICP-MS measurements from each digestion method were averaged and compared. These are shown in S6 (Deming regression of ICP/MS using different digestion methods). For the XACT, the standards used in calibrations were much higher than ambient concentrations and the calibration matrix differed from sample matrix (Indresand et al., 2013). Nevertheless, the results of the XACT comparison with ICP-MS in this study are comparable to those reported in other studies (Furger et al., 2017).



### 3.2.2    Comparison with ACSM at Marylebone Rd

The hourly values of S and Cl measured with the XACT were used to calculate hourly non-sea salt sulphate ($SO_4$), which was compared to the hourly sulphate (predominantly ammonium sulphate) which is non-refractory measured by the ACSM (Chang et al., 2011). The mean (median) concentrations were 2,600 (1,880) ng m$^{-3}$ and

2,000 (1,460) ng m$^{-3}$, respectively. The time series of these measurements is shown in Figure 5 and demonstrates the excellent temporal agreement, which is reflected by an $R^2$ of 0.93. The correlation resulted in a slope of 1.41 (95% CI 1.35-1.46) and an intercept of 53 (95% CI 13.4-93) ng m$^{-3}$. The larger non-sea salt $SO_4$ means/medians and slope >1 likely resulted from measuring different size fractions; $PM_{2.5}$ for the XACT vs $PM_1$ for the ACSM.

### 3.2.3    Comparison with URG

Hourly concentrations of water-soluble Cl, K and Ca measured by URG were compared to the hourly measured total Cl, K and Ca measured by the XACT. Furthermore, hourly measured water-soluble $SO_4$ (URG) was compared to hourly $SO_4$ calculated from the S measurement by the XACT instrument (Table 6, Figure 6). The XACT measured higher concentrations for all these components. The slopes were similar for the $SO_4$ (1.65) and

Cl (1.68) and slightly higher for Ca (1.89). Deming regression for K resulted in a very high slope (4.55) but this was likely the result of concentrations being close to the LOD for the URG, result was consistent with the findings presented by Beccaceci et al. (2015). The $R^2$ for Ca, Cl, K and $SO_4$ was 0.86, 0.93, 0.36 and 0.95, respectively.

The higher concentrations measured by the XACT relative to the URG was likely caused by the low water-

solubility of Cl, K, Ca and S containing minerals as well as the penetration efficiency of larger aerosols through the URG annular denuder (Beccaceci et al. 2015). The range of sources of these ions/elements resulted in variations in particle size and solubility and hence the relative response of the two instruments. When considering solubility, the larger slopes are associated with the least soluble compounds. In order of decreasing solubility (and increasing slope) $SO_4$ exists predominantly as $(NH_4)_2SO_4$ (solubility of 754 g/L in water), Cl is

principally from marine sources as NaCl (solubility of 359 g/L in water at 20ºC); Ca in the urban environment is typically from mineral or construction sources and is comprised of $CaCO_3$ and $CaSO_4·2H_2O$ (solubilities of 0.013 g/L and 2.55 g/L respectively) as well as calcium silicates (which are insoluble)(Dean and Lange, 1999). When considering particle size, the sources of aerosols containing Cl, K, Ca and S are often larger than $PM_{2.5}$ and may therefore be influenced by the reduced penetration efficiency of the URG annular denuder (Dick et al.,

1995; Visser et al., 2015b). Measured chemical composition of different size fractions at Marylebone Rd during winter and summer campaigns during 2012 and the percentage of the element in the $PM_{10-2.5}$ fraction can be used to highlight how these elements are distributed between the fine and coarse particle sizes: S 35%, K 57%, Ca 72%, and Cl 73%. This illustrates that a sampling bias, due to the penetration efficiency of the annular denuder may play a role in the difference between the URG and the XACT, however due to the additional

variation in solubility this is difficult to quantify.

### 3.3    Laboratory based filter analysis using the XACT

With a mean $R^2$ of 0.95 daily concentrations measured on the filter by the XACT compared well with the measurements made by the XACT when deployed in the field. The resulting regression slopes are compared





with those from the ICP-MS comparison (Figure 7). The small sample number ($N$ = 12) resulted in a higher uncertainty in the slopes but in general the slopes were comparable to those from the ICP-MS filter analyses. The intercepts for most elements were not significantly different from 0. The slopes for the elements Ba, Cl, Cr, Cu, Mn, Ni, Sr, V and Zn were not significantly different from the 1:1 line. For the elements As, Ca, Fe and Ti

the XACT measurements were lower when deployed in the field than when measuring the 24hr filter samples. Whereas for the elements K, Mo, Pb, S and Se online field measurements resulted in higher results than off-line filter measurements.. Full results can be found in the supplementary information.

Punching and subsequent filter analysis was found to be practically achievable, although time consuming, when compared to automated laboratory techniques.

**4    Conclusions**

This study was performed to evaluate the XACT 625 in the field and under laboratory conditions. In the field, the XACT was evaluated in three contrasting environments and compared to laboratory-based ICP-MS analysis as well as alternative high time resolution instrumentation. The XACT was found to be a highly reliable measurement instrument which showed an excellent correlation with standardised laboratory analysis (ICP-MS)

albeit with a slight overall positive bias (median 1.07). This was attributed to recovery rates from acid digestion and filter sampling. When compared to the alternative aerosol mass spectrometry and ion chromatography based high time resolution techniques, the XACT showed good temporal agreement but with a significant positive bias (median 1.68) compared to the ICP-MS; this was likely due to the differences in the size selection methodology employed by the different techniques as well as particle volatility and water solubility. However, these

differences in solubility and volatility could be utilised to provide information about different sources and their contributions; such as the difference between refractory sodium chloride and non-refractory ammonium chloride.

The laboratory experiment, which compared the XACT measurements of the elemental constituents of generated aerosols with the mass measured using a TEOM, proved to be a successful method for verifying the

response of the XACT over environmentally relevant elemental concentrations. The slopes were close to, and not significantly different from, unity (0.94 – 0.99). This suggests that the XACT accurately measures elemental ambient aerosol composition and that the positive bias, when compared to the ICP-MS measurements identified in the field experiments, was more likely due to filter artefacts and recovery rates following acid digestion. It further shows that generated aerosols can be used to calibrate the XACT to provide ongoing quality assurance

checks.

An ambient filter sampling analysis technique, using the XACT as a laboratory based instrument, was also evaluated. The concentrations measured on the sampled filter compared well with the in-situ XACT with median slopes of 1.07 and therefore comparable with the ICP-MS filter-based technique. This technique diversifies further the use of the XACT, especially if the instrument has downtime between campaigns. This

technique also allows a direct comparison of the XACT and other XRF systems using a filter sample.

Future work should include a repetition of the laboratory calibration using an overall lower range of standards and combining solutions in order to have a more complex particle composition. A standard reference material, either in solution or on filter should also be included in future calibration tests.



## 5 Acknowledgments

This study has been partly funded by the Welsh Government under contract C224/2015/2016 and by the UK
Department for Environment Food & Rural Affairs under contract AQ0740. It used equipment funded by the
Natural Environment Research Council Traffic Grant (NE/1007806/1) and by the UK Department for the

Environment and Rural Affairs Black Smoke and Heavy Metals Monitoring Networks.

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

**7    Figures**

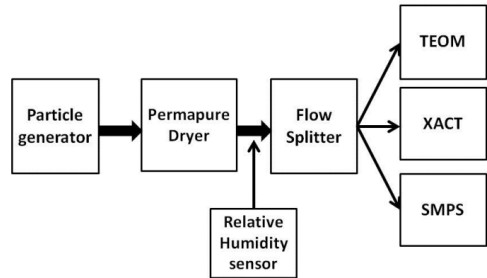

**Figure 1: Schematic of instrument set up during laboratory calibration**






**Figure 2: Deming regression of Cl (top left), K (top right), S (bottom left) and Zn (bottom right) mass concentrations measured with the XACT and calculated from TEOM mass measurements**

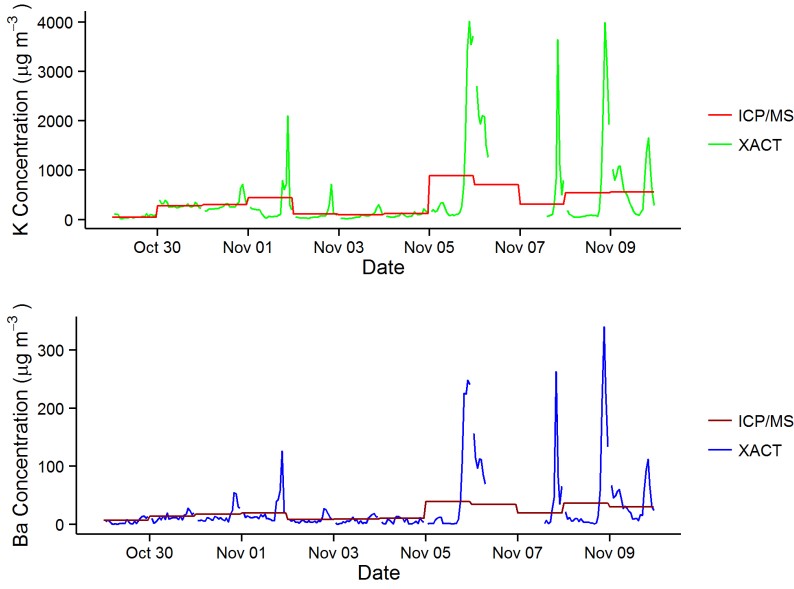

5    **Figure 3: Timeseries of K (top) and Ba (bottom) concentration (µg m⁻³) using hourly XACT and daily ICP-MS measurements**





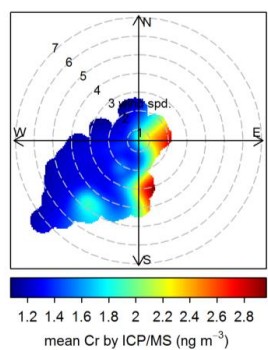 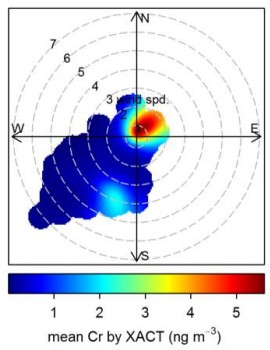

**Figure 4: Polar plot of the Cr concentrations (ng m$^{-3}$) in Pontardawe, Wales using daily ICP-MS measurements (left) and hourly XACT measurements (right)**

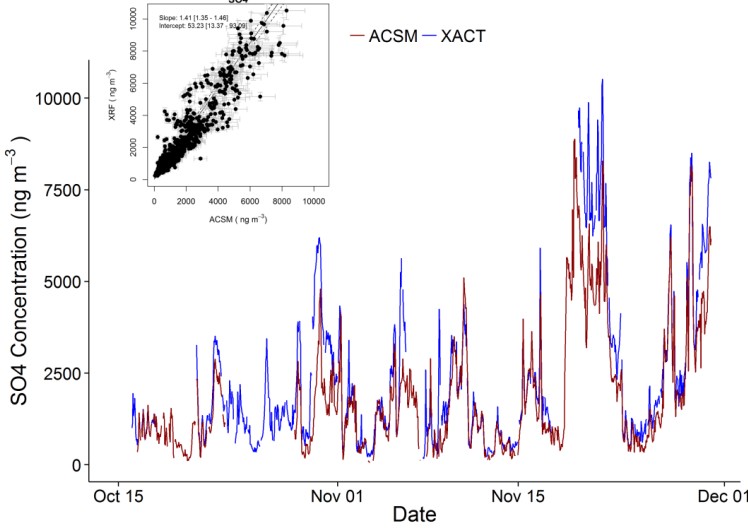

5    **Figure 5: Timeseries of non-sea salt SO$_4$ concentration (XACT, calculated) and non-refractory SO$_4$ (ACSM, measured) in ng m$^{-3}$**





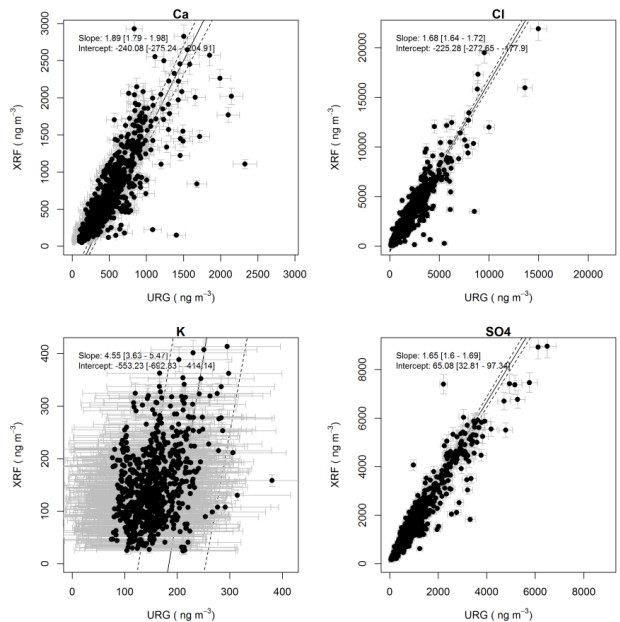

**Figure 6: Deming regression of water soluble Ca (top left), Cl (top right), K (bottom left) and SO₄ (bottom right) as measured by URG and Ca, Cl, K and calculated SO₄ (from elemental S) measured by XACT (ng m⁻³)**

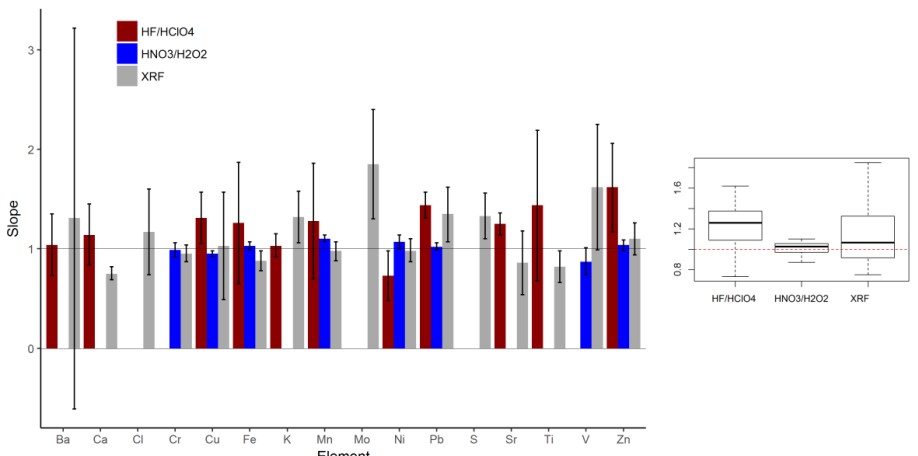

5    **Figure 7: Slope values (+95% confidence interval) of Deming regressions (XACT vs. ICP-MS (split in HF/HCLO₄ and HNO₃/H₂O₂ digestion) and XACT vs. XACT (filter)), split by element (left) and corresponding box-and-whisker plots split by method (right).**





## 8    Tables

**Table 1: Overview of sites and instrumentation used**

|  |  | Marylebone Road, London | Tawe Terrace, Pontardawe | Tinsley, Sheffield |
|---|---|---|---|---|
|  | PM$_{10}$ | PM$_{2.5}$ | PM$_{10}$ | PM$_{10}$ |
| XACT | 01-Jul-14 to 11-Mar-15 | 15-Oct-14 to 01-Dec-14 | 25-Nov-15 to 24-Dec-15 | 19-Jan-17 to 27-Mar- 17 |
| ACSM (PM$_1$) | n/a | 15-Oct-14 to 01-Dec-14 | n/a | n/a |
| URG | 07-Jan-15 to11-Mar-15 | n/a | n/a | n/a |
| Partisol | n/a | 15-Oct-14 to 01-Dec-14* | 25-Nov-15 to 24-Dec-15** | 19-Jan-17 to 27-Mar- 17**<br>17-Feb-17 to 10-Mar-17$^+$ |

Filters were digested using * HF/HClO$_4$ and ** HNO$_3$/H$_2$O$_2$
5    $^+$ Filters were analysed using the XACT in off-line mode

**Table 2: Maximum concentration in field campaigns (ng m$^{-3}$) and highest and lowest concentration used in calibration test**

| Field campaign | Concentration (ng m$^{-3}$) | | | |
|---|---|---|---|---|
|  | S | Cl | K | Zn |
| London Kerbsite (PM$_{10}$) | 3700 | 22000 | 470 | 310 |
| London Kerbsite* (PM$_{2.5}$) | 3500 | 4600 | 4000 | 370 |
| Wales Industrial (PM$_{10}$) | 8900 | 21000 | 1500 | 5500 |
| Sheffield Industrial (PM$_{10}$) | 4900 | 10000 | 1020 | 4900 |
| lowest standard | 2400 | 7200 | 8500 | 4900 |
| highest standard | 30000 | 35000 | 39000 | 20000 |





**Table 3: Overview of Marylebone Road, London measurements by XACT and ICP-MS (ng m⁻³); (* only 18 samples for XACT As)**

| | species | XACT (ng m⁻³) | | | | | | ICP/MS (ng m⁻³) | | | | | |
| --- | --- | --- | --- | --- | --- | --- | --- | --- | --- | --- | --- | --- | --- |
| | | mean | sd | med | min | max | LOD | mean | sd | med | min | max | LOD |
| Marylebone Road, London (n=19) | As* | 1.51 | 2.4 | 0.40 | 0.001 | 8.8 | 0.00020 | 0.97 | 1.02 | 0.53 | 0.049 | 4.0 | 0.099 |
| | Ba | 15.8 | 14.3 | 10.0 | 1.74 | 50 | 0.31 | 15.1 | 9.9 | 11.0 | 3.1 | 39 | 0.0166 |
| | Ca | 67 | 35 | 61 | 19.5 | 157 | 1.11 | 71 | 32 | 65 | 23 | 142 | 0.0166 |
| | Cd | 4.0 | 0.37 | 4.0 | 3.4 | 4.7 | 2.4 | 0.114 | 0.106 | 0.079 | 0.023 | 0.39 | 0.0046 |
| | Ce | 1.07 | 0.198 | 1.09 | 0.61 | 1.42 | 0.135 | 0.38 | 0.128 | 0.36 | 0.182 | 0.62 | 0.00030 |
| | Cl | 400 | 400 | 250 | 4.4 | 1180 | 2.1 | | | | | | |
| | Cr | 1.33 | 0.52 | 1.35 | 0.46 | 2.4 | 0.025 | | | | | | |
| | Cu | 21 | 7.3 | 21 | 6.5 | 35 | 0.29 | 16.5 | 6.6 | 14 | 3.7 | 29 | 0.187 |
| | Fe | 470 | 124 | 450 | 240 | 710 | 5.4 | 380 | 90 | 360 | 230 | 600 | 1.52 |
| | K | 230 | 230 | 103 | 59 | 870 | 7.9 | 230 | 230 | 110 | 48 | 890 | 8.1 |
| | Mn | 4.9 | 1.32 | 4.6 | 3.0 | 8.1 | 0.076 | 3.9 | 1.33 | 3.8 | 1.91 | 7.3 | 0.045 |
| | Mo | 0.64 | 0.109 | 0.62 | 0.46 | 0.97 | 0.40 | | | | | | |
| | Ni | 0.73 | 0.60 | 0.54 | 0.25 | 2.2 | 0.099 | 1.33 | 0.74 | 1.14 | 0.50 | 2.8 | 0.0044 |
| | Pb | 11.1 | 8.7 | 7.7 | 1.66 | 31 | 0.116 | 8.1 | 7.2 | 5.1 | 1.10 | 25 | 0.093 |
| | Pt | 0.177 | 0.010 | 0.175 | 0.161 | 0.20 | 0.078 | | | | | | |
| | S | 600 | 330 | 460 | 180 | 1600 | 3.3 | | | | | | |
| | Se | 0.177 | 0.169 | 0.112 | 0.070 | 0.77 | 0.031 | | | | | | |
| | Si | 110 | 70 | 85 | 72 | 340 | 65 | | | | | | |
| | Sr | 3.8 | 5.8 | 0.95 | 0.47 | 19 | 0.25 | 2.9 | 4.3 | 0.93 | 0.106 | 14.6 | 0.026 |
| | Ti | 4.4 | 2.7 | 3.4 | 1.80 | 12 | 0.158 | 2.9 | 2.5 | 1.97 | 0.28 | 9.1 | 0.067 |
| | V | 0.84 | 0.81 | 0.67 | 0.138 | 2.7 | 0.085 | | | | | | |
| | Zn | 27 | 16 | 21 | 5.2 | 57 | 0.195 | 22 | 10.9 | 17.7 | 7.5 | 39 | 1.43 |

5 **Table 4: Overview of Pontardawe, Wales measurements by XACT and ICP-MS (ng m⁻³)**

| | species | XACT (ng m⁻³) | | | | | | ICP/MS (ng m⁻³) | | | | | |
| --- | --- | --- | --- | --- | --- | --- | --- | --- | --- | --- | --- | --- | --- |
| | | mean | sd | med | min | max | LOD | mean | sd | med | min | max | LOD |
| Pontardawe, Wales (n=25) | As | 0.43 | 0.47 | 0.22 | 0.037 | 2.2 | 0.00020 | 0.23 | 0.31 | 0.081 | 0.030 | 1.12 | 0.037 |
| | Ba | 1.41 | 0.63 | 1.10 | 0.97 | 3.1 | 0.31 | | | | | | |
| | Ca | 191 | 109 | 155 | 50 | 510 | 1.11 | | | | | | |
| | Cd | 3.0 | 0.35 | 2.9 | 2.5 | 3.8 | 2.4 | 0.085 | 0.080 | 0.068 | 0.004 | 0.31 | 0.0110 |
| | Ce | 0.85 | 0.30 | 0.76 | 0.46 | 1.95 | 0.135 | | | | | | |
| | Cl | 5200 | 3000 | 5000 | 330 | 12700 | 2.1 | | | | | | |
| | Cr | 1.62 | 2.4 | 0.41 | 0.065 | 9.8 | 0.025 | 1.52 | 0.81 | 1.26 | 1.26 | 4.8 | 1.43 |
| | Cu | 3.8 | 2.2 | 3.9 | 0.67 | 8.9 | 0.29 | 4.0 | 2.2 | 3.7 | 0.63 | 9.1 | 0.099 |
| | Fe | 230 | 196 | 154 | 28 | 780 | 5.4 | 210 | 168 | 183 | 41 | 700 | 6.0 |
| | K | 154 | 60 | 138 | 83 | 340 | 7.9 | | | | | | |
| | Mn | 3.1 | 2.7 | 2.3 | 0.55 | 11.0 | 0.076 | 2.7 | 2.5 | 2.1 | 0.180 | 9.9 | 0.071 |
| | Mo | 1.15 | 2.1 | 0.58 | 0.45 | 10.2 | 0.40 | | | | | | |
| | Ni | 20 | 64 | 2.5 | 0.24 | 320 | 0.099 | 21 | 58 | 3.0 | 0.192 | 290 | 0.54 |
| | Pb | 3.7 | 4.3 | 2.6 | 0.29 | 21 | 0.12 | 2.9 | 3.5 | 1.99 | 0.140 | 16.6 | 0.22 |
| | Pt | 0.30 | 0.47 | 0.189 | 0.162 | 2.5 | 0.078 | | | | | | |
| | S | 530 | 240 | 450 | 196 | 1130 | 3.3 | | | | | | |
| | Se | 0.24 | 0.164 | 0.197 | 0.096 | 0.88 | 0.031 | 1.34 | 0.37 | 1.32 | 0.73 | 1.92 | 0.190 |
| | Si | 280 | 420 | 102 | 92 | 1820 | 65 | | | | | | |
| | Sr | 2.5 | 1.43 | 2.2 | 0.49 | 6.3 | 0.25 | | | | | | |
| | Ti | 8.7 | 15.4 | 2.8 | 0.61 | 65 | 0.158 | | | | | | |
| | V | 1.11 | 1.29 | 0.45 | 0.159 | 4.3 | 0.085 | 1.10 | 1.18 | 0.62 | 0.094 | 3.9 | 0.0160 |
| | Zn | 7.3 | 6.9 | 5.3 | 0.69 | 34 | 0.195 | 6.8 | 7.1 | 5.8 | 0.32 | 34 | 0.81 |





**Table 5: Overview of Tinsley, Sheffield measurements by XACT and ICP-MS (ng m⁻³)**

| | species | XACT (ng m⁻³) | | | | | | ICP/MS (ng m⁻³) | | | | | |
|---|---|---|---|---|---|---|---|---|---|---|---|---|---|
| | | mean | sd | med | min | max | LOD | mean | sd | med | min | max | LOD |
| | As | 2.9 | 4.8 | 1.35 | 0.035 | 33 | 0.00020 | 1.50 | 3.4 | 0.78 | 0.019 | 26 | 0.037 |
| | Ba | 2.6 | 3.6 | 1.75 | 0.98 | 28 | 0.31 | | | | | | |
| | Ca | 400 | 260 | 370 | 37 | 1100 | 1.11 | | | | | | |
| | Cd | 3.4 | 0.57 | 3.3 | 2.7 | 6.6 | 2.4 | 0.80 | 1.64 | 0.32 | 0.035 | 11.7 | 0.0110 |
| | Ce | 0.76 | 0.22 | 0.73 | 0.41 | 1.52 | 0.135 | | | | | | |
| | Cl | 1370 | 1100 | 1140 | 36 | 5100 | 2.1 | | | | | | |
| | Cr | 53 | 65 | 30 | 0.42 | 350 | 0.025 | 55 | 51 | 38 | 3.9 | 250 | 1.43 |
| Tinsley, Sheffield (n=60) | Cu | 17.5 | 11.0 | 14.6 | 2.3 | 47 | 0.29 | 19.3 | 12.5 | 16.0 | 2.6 | 56 | 0.099 |
| | Fe | 670 | 440 | 570 | 83 | 1950 | 5.4 | 680 | 420 | 580 | 92 | 1600 | 6.0 |
| | K | 138 | 92 | 108 | 17.0 | 420 | 7.9 | | | | | | |
| | Mn | 47 | 53 | 32 | 1.58 | 290 | 0.076 | 41 | 44 | 29 | 1.82 | 240 | 0.071 |
| | Mo | 15.1 | 24 | 7.0 | 0.65 | 130 | 0.40 | | | | | | |
| | Ni | 25 | 29 | 14.0 | 0.22 | 113 | 0.099 | 24 | 26 | 13.8 | 0.99 | 113 | 0.54 |
| | Pb | 22 | 23 | 13.1 | 1.33 | 125 | 0.116 | 22 | 22 | 11.8 | 1.21 | 111 | 0.22 |
| | Pt | 0.186 | 0.017 | 0.185 | 0.166 | 0.28 | 0.078 | | | | | | |
| | S | 780 | 670 | 550 | 126 | 3400 | 3.3 | | | | | | |
| | Se | 0.93 | 1.30 | 0.31 | 0.075 | 5.5 | 0.031 | 1.83 | 1.62 | 0.94 | 0.26 | 6.2 | 0.190 |
| | Si | 210 | 150 | 164 | 71 | 780 | 65 | | | | | | |
| | Sr | 1.15 | 0.68 | 1.10 | 0.41 | 3.6 | 0.25 | | | | | | |
| | Ti | 23 | 36 | 14.3 | 1.42 | 220 | 0.158 | | | | | | |
| | V | 1.16 | 2.0 | 0.60 | 0.179 | 12.9 | 0.085 | 1.45 | 1.47 | 1.02 | 0.171 | 9.6 | 0.0160 |
| | Zn | 100 | 120 | 58 | 4.5 | 620 | 0.195 | 101 | 117 | 56 | 3.8 | 610 | 0.81 |

**Table 6: Overview of Marylebone Road, London SO₄ measurements in PM₂.₅ by XACT (SO₄\* calculated as non-sea salt SO₄ using S and Cl measurements) and ACSM (ng m⁻³); and SO₄, K, Cl, Ca measurements in PM₁₀ by XACT (SO₄\*\* calculated as predicted SO₄ using S measurements) and URG (ng m⁻³)**

| species | n | XACT (ng m⁻³) | | | | | | ACSM (ng m⁻³) | | | | | |
|---|---|---|---|---|---|---|---|---|---|---|---|---|---|
| | | mean | sd | med | min | max | LOD | mean | sd | med | min | max | LOD |
| SO₄\* | 737 | 2600 | 2200 | 1880 | 240 | 10500 | n/a | 2000 | 1700 | 1460 | 58 | 8300 | 35 |

| species | n | XACT (ng m⁻³) | | | | | | URG (ng m⁻³) | | | | | |
|---|---|---|---|---|---|---|---|---|---|---|---|---|---|
| | | mean | sd | med | min | max | LOD | mean | sd | med | min | max | LOD |
| SO₄\*\* | 1045 | 1750 | 1210 | 1450 | 164 | 9000 | n/a | 1040 | 810 | 810 | 54 | 6500 | 100 |
| K | 776 | 145 | 69 | 133 | 24 | 410 | 6.2 | 154 | 42 | 150 | 75 | 380 | 100 |
| Cl | 1045 | 2700 | 2400 | 2100 | 42 | 22000 | 9 | 1790 | 1530 | 1370 | 132 | 15000 | 100 |
| Ca | 996 | 590 | 490 | 430 | 49 | 2900 | 3.3 | 440 | 300 | 360 | 97 | 2300 | 100 |



**Table 7: Deming regression results and coefficient of determination for XACT comparison with ICP-MS, separated by HF/HClO$_4$ and HNO$_3$/H$_2$O$_2$ digestions**

| Element | HF/HClO$_4$ | | | HNO$_3$/H$_2$O$_2$ | | |
|---|---|---|---|---|---|---|
| | **Slope** | **Intercept** | **R$^2$** | **Slope** | **Intercept** | **R$^2$** |
| As | 2.0 (1.49-2.6) | -0.33 (-0.65-0) | 0.95 | 3.8 (1.90-5.7) | -0.23 (-0.49-0.020) | 0.90 |
| Ba | 1.04 (0.73-1.35) | -1.50 (-4.8-1.79) | 0.98 | | | |
| Ca | 1.14 (0.84-1.45) | -9.2 (-31-13) | 0.70 | | | |
| Cr | | | | 0.99 (0.92-1.06) | -1.70 (-2.6--0.79) | 0.95 |
| Cu | 1.31 (1.05-1.57) | 0.29 (-3.1-3.7) | 0.93 | 0.95 (0.92-0.98) | -0.03 (-0.22-0.17) | 0.89 |
| Fe | 1.26 (0.65-1.87) | -1.29 (-220-210) | 0.89 | 1.03 (0.99-1.07) | -10 (-18.19--2.0) | 0.96 |
| K | 1.03 (0.92-1.15) | -1.23 (-14.83-12.37) | 0.96 | | | |
| Mn | 1.28 (0.70-1.86) | 0.050 (-1.97-2.1) | 0.92 | 1.10 (1.07-1.14) | 0.17 (0.020-0.32) | 0.99 |
| Ni | 0.73 (0.48-0.98) | -0.20 (-0.45-0.05) | 0.67 | 1.07 (1.00-1.14) | -1.21 (-1.64-0.77) | 0.99 |
| Pb | 1.44 (1.31-1.57) | 0.140 (-0.37-0.65) | 1.00 | 1.02 (0.99-1.06) | 0.36 (0.10-0.61) | 0.99 |
| Se | | | | 0.83 (0.73-0.94) | -0.45 (-0.57--0.33) | 0.67 |
| Sr | 1.25 (1.14-1.36) | -0.0100 (-0.19-0.17) | 1.00 | | | |
| Ti | 1.44 (0.68-2.2) | 0.91 (-0.42-2.2) | 0.72 | | | |
| V | | | | 0.87 (0.74-1.01) | -0.130 (-0.22--0.04) | 0.89 |
| Zn | 1.62 (1.17-2.1) | -4.4 (-13.15-4.5) | 0.50 | 1.04 (0.98-1.09) | 0.37 (-0.58-1.31) | 0.94 |

5    **9    Copyright statement**

**10    Data Availability**

ICP-MS measurements on filters made at Pontardawe and Sheffield are available from https://uk-air.defra.gov.uk/data/metals-data. Additional datasets are available upon request to the corresponding author.

15    **11    Supplement link**

The Supplement related to this article is available online at <enter link when available>



## 12    Author Contribution

Anja H. Tremper - experiment design (XACT filter analysis), field and laboratory experiments, data ratification and analysis, manuscript preparation with contributions from all co-authors

Anna Font - field experiments and data analysis

Max Priestman - field experiments and ACSM data ratification

Samera H Hamad - field experiments (Marylebone Road)

Tsai-Chia Chung - laboratory experiments (calibration test of Cl, K and S)

Ari Pribadi - laboratory experiments (filter analysis)

Richard J. C. Brown - uncertainty calculations

Sharon L. Goddard - filter analysis (Wales/Sheffield)

Nathalie Grassineau - filter digestion (Marylebone Road)

Krag Petterson - technical input for XACT

Frank J. Kelly – manuscript review

David C. Green - experiment design (laboratory calibration test), field and laboratory experiments, uncertainty

calculations, manuscript overview

## 13    Competing interests

Krag Petterson is employed by Cooper Environmental Services, the manufacturer of the instrument, and had input into the manuscript preparation from a technical perspective.

## 14    Disclaimer

This manuscript has not been published and is not under consideration for publication elsewhere.