# Peer review of "Field and laboratory evaluation of a high time resolution xray fluorescence instrument for determining the elemental composition of ambient aerosols"

_Atmospheric Measurement Techniques, 2017_

## Referee Comment (RC1) · Anonymous Referee #2 · 4 Jan 2018

The paper refers to a very important issue in the characterization of an aerosol sampling/analyzing device, namely, its analytical accuracy. Therefore, the work is relevant. There is a careful design of the methods used to validate the quantitative results. They are carefully explained and presented. It is remarkable that several analytical techniques are used to carry out the comparisons. I could find no scientific errors or misleading discussions. The conclusions actually point out to the results obtained in the text. Also, proposals to extend (and improve) the results are given. In short, I consider the paper should be accepted for publication. Nevertheless, I would like to suggest

a few (minor) corrections and additions. 1. In section 2.1, I recommend adding a few lines mentioning the x-ray source and its operating conditions employed, as well as the detector type and characteristics (resolution, in particular). This may help the reader to better understand the results given in this manuscript. Also, the software and atomic databases used to analyze the x-ray spectra. 2. The aforementioned information might be useful to understand several of the apparently incorrect results, like the As overestimation. For instance, the As Kalpha x-ray peak overlaps the Pb Lalpha peak. Therefore, it is very important to carefully integrate and correct both peaks using the corresponding beta lines. This may be the reason of the extremely high overestimation in the measured As concentrations. Although it is not explained in depth, Se is another element with possible problems in quantification (see Fig. S7). 3. The plots in Figs. 2 and 6 need a larger lettering to facilitate reading. Moreover, instead of using thousands of nanograms, possibly using micrograms is easier. 4. Only as minor but important corrections in writing style, expressions like the one given in page 6, line 13, "75-650 nm" must be written as "75 nm to 650 nm," according to the International System style rules (please, read the official document in the IBPM web site). This must be corrected in all the manuscript. Similar changes must be made when writing quantities (like those in page 6, lines 35 and 38), where a space between the numerical value and unit symbol is missing. 5. Also referring to the official document of the SI, the units "ppb" must be avoided, because of the different meaning of "billion" in diverse countries and languages.

---

## Referee Comment (RC2) · Anonymous Referee #1 · 9 Jan 2018

**General**

The manuscript presents a comprehensive lab and field evaluation of the CES Xact 625 XRF spectrometer. Various alternative methods with different time resolutions and particle sizes are compared with the XRF measurements, with generally good agreement to XRF and between methods. Explanations for deviations are given. The manuscript adds to the literature on quality assessment of the Xact online spectrometer, an instrument with great potential for monitoring environmental metals and other elements in airborne particles. The study is more comprehensive than previous studies that

compare those methods, although some questions with Xact data quality remain.

The structure of the manuscript, the results and the presentation of the material are good. The data has been analyzed and presented with care. The language is impeccable. The topic is relevant and well worth publication in AMT.

**Major comments**

The study describes intercomparisons of time series of metal concentrations measured with the Xact and with other techniques. The statistical work horse is Deming regression which assigns individual measurement uncertainties to each technique. This approach is straightforward and has been applied in other studies. Comparisons are made between Xact and ICP-MS (with different digestions), ACSM, URG, and XRF (measuring filters with the Xact itself). Data were collected in field studies and in laboratory experiments.

The field studies show good regressions with a slightly positive bias of the Xact vs. ICP-MS, in agreement with previous studies. It is nicely demonstrated that the slope depends on the digestion method used for ICP-MS, and an overestimation of the XRF values compared to the ICP-MS values may turn into an underestimation when using the other digestion method. This makes a generalization of regression results (which technique is best or better?) rather difficult. Comparisons with ACSM and URG suffer from the different size classes sampled (PM2.5, PM1) and the different particle characteristics (non-sea salt, non-refractory). A comparison then requires additional assumptions to bring the values into closer agreement. This is discussed by the autors, but it also makes the comparisons more qualitative than quantitative. Interesting is the comparison between Xact measurements and filters analyzed with the Xact (Table S1). Slopes ranging from 0.8 to 1.85 with R2 values > 0.9 (excluding the extreme cases of As and Se) might indicate more serious issues with the calibration of the Xact or with the spectral deconvolution algorithm, even though the number of samples is only 12, and slopes of 1.31 (Ba) and 1.62 (V) are not significantly different from unity,
according to the authors. Here I would like some comment from the authors.

The laboratory experiment regressions in Fig. S2 show very good agreement between Xact and TEOM, except for some outliers for S. The concentration maxima are extremely high compared to typical ambient concentrations. Except Zn, all elements should be prone to self-absorption effects in XRF analysis when the deposited layer becomes too thick, but no such effect can be seen in the regressions. Are self-absorption effects so well compensated by the Xact software? It would be helpful to add information on particle sizes (as measured with the SMPS) and/or deposit thickness to understand why XRF self-absorption effects do not show up in the graphs.

**Minor comments**

P7L1: Middlebrook instead of Middlebook P7L34-36: Strange sentence. P8L11: remove one '.' after Table 3. P12L7: remove one '.' after filter measurements. P21 Table 7: Check the arrangement of rows carefully. P17 Fig. 7 and Figs. S3-S5: It might make the graphs more consistent when the coloring of all figures were in agreement. I suggest to color the dots in the Figures S3-S5 in red (HF/HCIO4), and blue (HNO3/H2O2), to correspond to the colors in Fig. 7.

AMTD

---

## Referee Comment (RC3) · Anonymous Referee #5 · 10 Jan 2018

The new version of the article has been greatly improved. Some modification have still to be done before publication: 1) In section 3.1 the authors should explicitly cite the fact that in XRF there are neither self-absorption problems for the medium-high Z elements nor matrix effects. There can be self-absorption effects only for low Z elements (the only one, which can be affected in their comparison, is Si) even with samples with high loading unless the deposit on a very small area. Those effects depend mostly on the absorption within the single particle therefore they are present also in the samples prepared by the authors. The use of self-made standard can be useful, but I do not

see any problem in the use of commercial standards as it is done in many laboratories which routinely use XRF for aerosol analysis. 2) Again what is reported at the end of section 3.2.1 page 11 lines 12-13 is not correct (same comment as above) 3) Section 3.2.2: the use of Cl to calculate non sea-salt sulphate can give a strong overestimation of that component due to the possible volatilization of Cl in case of aged sea-salt as reported in many works regarding also the sites analyzed by the authors. Normally Na is used. The authors must make a comment about this 4) All the information reported by the authors are interesting and better explained in this new version. However, in my opinion, it cannot be neglected that the best way to assess the performances of the XACT spectrometer would have been to use a standard aerosol sampler (like one of those used by the authors), the proper collection filters (e.g. Teflon or polycarbonate filters) and XRF analysis of the collected filters. The authors should make an explicit comment about this.

---

## Referee Comment (RC4) · Anonymous Referee #3 · 10 Jan 2018

The paper "Field and laboratory evaluation of a high time resolution x-ray fluorescence instrument for determining the elemental composition of ambient aerosols" by Tremper et al. is a comprehensive study aiming at providing laboratory and in-field information on the performance of the X-ACT 625 instrument. The paper is well structured, data presentation is adequate and well commented. X-ACT is an innovative instrument and improving its characterization is important for the scientific community. Thus, in my opinion the paper is of interest for publication in AMT and publication can occur after few revisions are performed.

[Figure]

Major concerns:

P10L18: why is Cd not mentioned? Cd is the element providing the strongest differences in all cases, but its discussion is completely missed in the text. Please add comments about it, or give explanation why it should be rejected. In this case, please remove it throughout the manuscript.

P11L30-35: Parallel sampling of PM10 and PM2.5 is needed to determine PM2.5/PM10 element ratios. Sampling different aerosol size fractions in different seasons and using ratios to separate fine and coarse contributions by elements is misleading. Please remove.

X-ACT is an on-line device providing elemental composition of atmospheric aerosol. Other instruments (e.g. streaker sampler, rotating drum impactors) can provide high time-resolved measurements by off-line analyses performed at accelerator facilities (e.g. by Particle-Induced X-Ray Fluorescence or Synchrotron Radiation XRF). I think a comparison with such measurements should be mentioned as a perspective.

Minor concerns: P2L2: please evidence that modelling approaches (and not only measurements of aerosol chemical composition) are needed to gain information on aerosol sources

P3L2: please add ion chromatography for inorganic ions (as it is cited in the following and applied in the paper)

P3L23: "sample a narrower range of components". Please change "sample" with "measure" (instruments sample what is in air but are not always able to quantify)

P5L3: wrong formula for Ammonium sulphate (cfr. P5L13 where it is correct)

P6L6: "where source contributions may be assumed based on one of these measurement techniques". I guess the authors refer to receptor modelling approaches for source apportionment. Please change into "where source contributions may be estimated by receptor modelling using measurements of chemical components as input"

(the measurement techniques do not provide source contributions, but quantify chemical components)

P7L3-5: obscure. Please add some explanation

P8L7: 3 or sqrt(3)?

P8L12&15: what is k?

P8L27: Are the authors referring to statistical significance?

P9L1: "dominated by fireworks activity (Oct-Dec 2014)". Do the authors mean that fireworks were the main source impacting the area in those 3 months? If not, please rephrase.

P9L7-19: First of all, suitable references to fireworks tracers in aerosol are missing. Secondly, high time resolved measurements of elemental composition during fireworks has already been presented in the literature. In one case, they were also exploited for source apportionment by receptor models (Vecchi et al., 2008. DOI: 10.1016/j.atmosenv.2007.10.047)

P9L22&34: "mean concentrations". Please change into "mean elemental concentrations"

P10L1&P11L2: how was non-sea sulphate calculated? Please describe or add suitable references. Furthermore, provide references for this choice in the comparison.

P10L28: "filter artefacts"? What do the authors refer to? Sampling artefacts or something else? Please, clarify.

Typos

P2L31: "implementing" instead of "implement"

P7L1 vs P7L2: Middlebook or Middlebrook?

P10L30: change "extends" into "extents"

P11L7: ".." Change into "."

---

## Referee Comment (RC5) · Anonymous Referee #4 · 15 Jan 2018

Overall the manuscript presents relevant results in a comprehensive and well-organized manner. So the publication in AMT is recommended. Still some conclusions need to be revised or softened, or better supported. Please see the comments below.

- Abstract. State which elements are compared with ICP and which with 'other high time resolution measurements', because the differences in slopes (median 1.07 vs 1.41-4.6) may be due to the different elements being assessed rather than to differences between techniques?

[Figure]

- Page 3. Line 4. The positive and negative sampling artefacts are true for some species but not for others, e.g. metals concentrations determined on filter samples by digestion+ICP do not suffer from sampling artefacts.

- Page 3. Lines 34-36. Furger et al. (2017) used both ICP-OES and ICP-MS, not only ICP-MS for the list of elements reported in this manuscript.

- Page 6. Lines 39-40 and page 7, line 1. Wasn't the RIE for ammonium calculated from the calibration with ammonium nitrate? And hence only the RIE for sulphate calculated from calibration with ammonium sulphate? Please correct if necessary.

- Page 9, lines 22-23. Please modify the sentence. As written now it seems you are still taking about the Ni, and according to Table 4, Ni concentration is reported to be 20 and min and max 0.24 and 320. Hence, from Table 4, one can see that 0.24-5200 is the range of mean concentrations for all the species analysed, but from the text is not clear at all.

- Page 9, line 25. The reason for high Cl concentration is not only the PM10 head, as at Tinsley the head was also PM10 but Cl is not so high. I guess the proximity to ocean played a role here.

- Page 10, line 2. With hourly concentration ranging. . .? Is it hourly? Or?

- Page 10, line 8. They are not ICP-MS digestion methods, they are digestion methods. The ICP-MS is used afterwards. Please re-write.

- It is not so clear that the differences XACT vs ICP can be attributed to the recovery rates in the digestion processes prior to ICP analysis. Please amend through the manuscript this explanation (especially conclusions, page 12 lines 15 and following, line 28). One needs to asses this statement based on the individual elements. If that statement was true, the elements with the lowest recovery rates would have highest slopes, but this is not the case e.g. recovery for Ni (HF/HClO4) is 87- In the conclusions, again, please make sure you don not attribute the difference between XACT and ICP to filter artefacts when you discuss elements that do not suffer from filter artefacts (page 12, line 28, among others).

- Conclusions, page 12, lines 17-18. If the sampling size was different and it is true

that the size range 1-2-5 um has so much sulphate, then the 1.68 is not a bias. The 1.68 is not a bias but would have actual meaning. Please re-phrase.

Technical corrections:
- Page 7, line 8. Shouldn't it say ". . .described in Beccaceci et al. (2015)"?
- Page 8, line 11. Remove 1 point after "Table 3".
- Page 8, line 13. A space is missing before "For".

---

## Author Comment (AC1) · 16 Mar 2018

Please find below the author's response to the interactive discussion comments from the anonymous referee.

Referee comments: RC

Author's response given below individual referee comments

RC - General The manuscript presents a comprehensive lab and field evaluation of the

CES Xact 625 XRF spectrometer. Various alternative methods with different time resolutions and particle sizes are compared with the XRF measurements, with generally good agreement to XRF and between methods. Explanations for deviations are given. The manuscript adds to the literature on quality assessment of the Xact online spectrometer, an instrument with great potential for monitoring environmental metals and other elements in airborne particles. The study is more comprehensive than previous studies that compare those methods, although some questions with Xact data quality remain. The structure of the manuscript, the results and the presentation of the material are good. The data has been analyzed and presented with care. The language is impeccable. The topic is relevant and well worth publication in AMT.

Author's response: We would like to thank the anonymous referee for the positive feedback and the suggestions to our manuscript. Please find the answers to the individual comments below.

RC - Major comments The study describes intercomparisons of time series of metal concentrations measured with the Xact and with other techniques. The statistical work horse is Deming regression which assigns individual measurement uncertainties to each technique. This approach is straightforward and has been applied in other studies. Comparisons are made between Xact and ICP-MS (with different digestions), ACSM, URG, and XRF (measuring filters with the Xact itself). Data were collected in field studies and in laboratory experiments. The field studies show good regressions with a slightly positive bias of the Xact vs. ICP-MS, in agreement with previous studies. It is nicely demonstrated that the slope depends on the digestion method used for ICP-MS, and an overestimation of the XRF values compared to the ICP-MS values may turn into an underestimation when using the other digestion method. This makes a generalization of regression results (which technique is best or better?) rather difficult. Comparisons with ACSM and URG suffer from the different size classes sampled (PM2.5, PM1) and the different particle characteristics (non-sea salt, non-refractory). A comparison then requires additional assumptions to bring the values into closer agreement. This is discussed by the authors, but it also makes the comparisons more qualitative than quantitative. Interesting is the comparison between Xact measurements and filters analyzed with the Xact (Table S1). Slopes ranging from 0.8 to 1.85 with R2 values > 0.9 (excluding the extreme cases of As and Se) might indicate more serious issues with the calibration of the Xact or with the spectral deconvolution algorithm, even though the number of samples is only 12, and slopes of 1.31 (Ba) and 1.62 (V) are not significantly different from unity, according to the authors. Here I would like some comment from the authors.

Author's response: The authors do not believe that the differences in slope indicate an issue with the calibration of the XACT. The reasons for the discrepancies can more likely be found due to the small sample size of this pilot study, the authors feel that the analysis method, including filter substrate and punching technique could be optimized. Nevertheless, every effort was made to make this as robust as possible - blank filters were analysed and used to correct for the filter background and to calculate the limit of detection when the XACT was used with filters on a 15 min analysis cycle. In comparison to the online XACT measurements which were made on an hourly basis, the LOD of the filter method was much higher. This was likely due to the shorter analysis time, the different area of deposit on a filter sample and the filter material (Zeflour has a density of around 16 mg cm-2 whereas our filter tape has a density of about 2 mg cm-2; the mass density of the filter material will impact the XRF detection limit by the ratio of the square root of the mass density. Further the fitting routine in the deconvolution software is optimised for the filter tape used and is another potential area of optimisation if this approach were to be pursued.

To make this clearer in the manuscript we have added the following to the "Materials and Methods" section 2.4 (P7L30):

"For quality assurance purposes, field and laboratory filter blanks were analysed and used to correct for the filter background. The blank measurements were also used to calculate the limit of detection for this method."

We have also expanded the results section 3.3 (P12L18) to include the above discussion:

"Reasons for the discrepancies in the slopes may be caused by the difference between the filter material and analysis time used for the filter samples (Zeflour, 15 min) in comparison to the online method (proprietary PTFE tape, 1 hr). Additionally the fitting routine used in the deconvolution software is optimised for the filter tape used and might also contribute to the observed differences."

We have further added a comment in the conclusions section 4 (13L18): "Further, to develop the filter analysis method using the XACT and piloted in this study, different filter materials should be tested and the deconvolution approach optimised if necessary."

RC - The laboratory experiment regressions in Fig. S2 show very good agreement between Xact and TEOM, except for some outliers for S. The concentration maxima are extremely high compared to typical ambient concentrations. Except Zn, all elements should be prone to self-absorption effects in XRF analysis when the deposited layer becomes too thick, but no such effect can be seen in the regressions. Are self-absorption effects so well compensated by the Xact software? It would be helpful to add information on particle sizes (as measured with the SMPS) and/or deposit thickness to understand why XRF self-absorption effects do not show up in the graphs.

Author's response: Sample self absorption depends mostly on the thickness of the deposit and its composition and even at these high concentration levels sample absorption effects contribute a relatively small amount to the overall analysis result. Sample self absorption was calculated for S and even at the highest concentration self absorption effects are less than 1%. Please also see response to referee 5.

Nevertheless the authors have changed the sentence in section 3.1: "All calibrations resulted in a linear relationship between the mass calculated using TEOM mass concentrations and measured by the XACT for the standard range used."

It now reads (P8L32): "All calibrations resulted in a linear relationship between the mass calculated using TEOM mass concentrations and measured by the XACT for the standard range used. Sample self absorption effects were calculated to be <1% for the maximum concentration of S (the lightest element used) and therefore insignificant in the use of this instrument."

Minor comments RC - P7L1: Middlebrook instead of Middlebook

Author's response: The spelling of "Middlebrook" has been corrected on P7L4: "The collection efficiency was calculated using the Middlebrook parameterisation (Middlebrook et al., 2012)..."

RC - P7L34-36: Strange sentence.

Author's response: Changed the sentence "Including values below the LOD had the advantage of being able to include daily XACT mean concentration was calculated from hourly concentrations that might have been lost if data below LOD was excluded and the daily data capture was not met."

to (P8L3) "By including values below the LOD it was possible to calculate daily XACT mean concentrations, which might have been lost if data below the LOD had been excluded and the daily data capture had not been met."

RC - P8L11: remove one '.' after Table 3.

Author's response: Removed the duplicated "." On P8L18 "...; these are shown in Table 3. For the ACSM..."

RC - P12L7: remove one '.' after filter measurements.

Author's response: Removed the duplicated "." On P12L18 "...resulted in higher results than off-line filter measurements."

RC - P21 Table 7: Check the arrangement of rows carefully.

Author's response: Formatted Table 7 on P22 using Font "Times New Roman" and Font Size 8 as used in the other tables; this corrected the formatting problem.

RC - P17 Fig. 7 and Figs. S3-S5: It might make the graphs more consistent when the coloring of all figures were in agreement. I suggest to color the dots in the Figures S3-S5 in red (HF/HClO4), and blue (HNO3/H2O2), to correspond to the colors in Fig. 7.

Author's response: To avoid confusion with the different colouring, the authors decided to keep the colouring of Fig. 7 as it is, but change Figs. S3-S5 and S7 to use black dots for all graphs.

Please also note the supplement to this comment:
https://www.atmos-meas-tech-discuss.net/amt-2017-363/amt-2017-363-AC1-supplement.pdf

---

## Author Comment (AC2) · 16 Mar 2018

Please find below the author's response to the interactive discussion comments from the anonymous referee.

Referee comments: RC

Author's response given below individual referee comments

RC - The paper refers to a very important issue in the characterization of an aerosol

sampling/analyzing device, namely, its analytical accuracy. Therefore, the work is relevant. There is a careful design of the methods used to validate the quantitative results. They are carefully explained and presented. It is remarkable that several analytical techniques are used to carry out the comparisons. I could find no scientific errors or misleading discussions. The conclusions actually point out to the results obtained in the text. Also, proposals to extend (and improve) the results are given. In short, I consider the paper should be accepted for publication. Nevertheless, I would like to suggest a few (minor) corrections and additions.

Author's response: We would like to thank the anonymous referee for the very positive feedback and the suggestions to our manuscript. Please find the answers to the individual comments below. The original comments are written in black and the author's reply and changes to the manuscript are coloured in blue/green, respectively.

RC - 1. In section 2.1, I recommend adding a few lines mentioning the x-ray source and its operating conditions employed, as well as the detector type and characteristics (resolution, in particular). This may help the reader to better understand the results given in this manuscript. Also, the software and atomic databases used to analyze the x-ray spectra.

Author's response: The authors have added further information on the XRF analysis to section 2.1. to help the reader understand the results.

The section has changed from: "The instrument measures 24 elements between Silicon and Uranium at a time resolution between 15 minutes and four hours using ED-XRF. The size fraction of the PM sample collected onto the Teflon filter tape depends on the size selective inlet chosen. The instrument samples with a volumetric flow rate of 1 m3 h-1 through an inlet tube heated to 45 ËŽC when the ambient relative humidity (RH) exceeds 45% to avoid water depositing on the tape. Sampling and analysis is performed continuously and simultaneously, except for the time required to advance the filter tape (âĹij20 s) from the sample to the analysis position. Daily automated quality assurance checks are performed every night at midnight and consist of an energy alignment (an energy calibration using a copper rod, inserted into the analysis area); and upscale measurement to monitor the stability of the instrument response (for Cd, Cr and Pb); and a flow check through an independent mass flow sensor. Additional quality assurance checks employed here included flow calibrations, regular external standard checks, field blanks performed using a HEPA filter as well as tape blanks before and after each tape change."

It now reads (added text is underlined, P4L30 onwards): "The instrument measures 24 elements between Silicon and Uranium at a time resolution between 15 minutes and four hours using ED-XRF. The size fraction of the PM sample collected onto the Teflon filter tape depends on the size selective inlet chosen. The instrument samples with a volumetric flow rate of 1 m3 h-1 through an inlet tube heated to 45 ËŽC when the ambient relative humidity (RH) exceeds 45% to avoid water depositing on the tape. Sampling and analysis is performed continuously and simultaneously, except for the time required to advance the filter tape (âĹij20 s) from the sample to the analysis position. During the analysis, the sample is excited using an x-ray source (Rhodium anode, 50 kV, 50 Watt) in three successive energy conditions, which target three different suites of elements. The resulting x-ray fluorescence is measured with a silicon drift detector and the spectra are analysed using a proprietary spectral analysis package which takes into account all peaks associated with a given element. Daily automated quality assurance checks are performed every night at midnight and consist of an energy alignment (an energy calibration using a copper rod, inserted into the analysis area); and upscale measurement to monitor the stability of the instrument response (for Cd, Cr and Pb); and a flow check through an independent mass flow sensor. Additional quality assurance checks employed here included flow calibrations, regular external standard checks, field blanks performed using a HEPA filter as well as tape blanks before and after each tape change."

RC - 2. The aforementioned information might be useful to understand several of the

apparently incorrect results, like the As overestimation. For instance, the As Kalpha x-ray peak overlaps the Pb Lalpha peak. Therefore, it is very important to carefully integrate and correct both peaks using the corresponding beta lines. This may be the reason of the extremely high overestimation in the measured As concentrations. Although it is not explained in depth, Se is another element with possible problems in quantification (see Fig. S7).

Author's response: The authors added text relating to the spectral peak fitting process, which takes into account all peaks associated with a given element, in the answer to the comment above. Further the equipment supplier has made us aware of a US-EPA verification report (US-EPA, 2012) which analysed Se and found an excellent agreement (R2= 0.926) and a brief summary of this has been added to the introduction Thus the paragraph was changed from: "Despite these limitations, the XACT is unique in measuring elements automatically using energy dispersive XRF (ED-XRF) and has been successfully evaluated in a number of field studies (Furger et al., 2017; Park et al., 2014). Park et al. (2014) found..."

It now reads (P3L33 onwards): "Despite these limitations, the XACT is unique in measuring elements automatically using energy dispersive XRF (ED-XRF) and has been successfully evaluated in a number of field studies (Furger et al., 2017; Park et al., 2014; US-EPA, 2012). In a verification test carried out by the US-EPA (2012) measurements of Ca, Cu, Mn, Pb, Se and Zn by the XACT were compared to filter based measurements (filters analysed using ICP-MS). This verification test showed that the daily average Xact 625 results were highly correlated and in close quantitative agreement with ICP-MS analysis results for the six metals, except Cu, which was close to the detection limit in many cases. Park et al. (2014) found..." The reference was added to the reference list: "US-EPA: Environmental Technology Verification Report. Cooper Environmental Services LLC Xact 625 Particulate Metals Monitor, Report no. EPA/600/R-12/680. Agency, U. S. E. P. A. (Ed.), U.S. Environmental Protection Agency, Cincinnati, OH 45268, 2012."

RC - 3. The plots in Figs. 2 and 6 need a larger lettering to facilitate reading. Moreover, instead of using thousands of nanograms, possibly using micrograms is easier.

Author's response: Increased the overall figure size of Figs. 2 and 6 to facilitate reading as increasing the font size of the equations would have interfered with the lines/graph, especially in Fig.6. For consistency throughout the manuscript it was decided to keep ng m-3 for all graphs and tables.

RC - 4. Only as minor but important corrections in writing style, expressions like the one given in page 6, line 13, "75-650 nm" must be written as "75 nm to 650 nm," according to the International System style rules (please, read the official document in the IBPM web site). This must be corrected in all the manuscript. Similar changes must be made when writing quantities (like those in page 6, lines 35 and 38), where a space between the numerical value and unit symbol is missing.

Author's response: Changed the manuscripts according to the International System of Units rules.

RC - 5. Also referring to the official document of the SI, the units "ppb" must be avoided, because of the different meaning of "billion" in diverse countries and languages.

Author's response: Changed "(18.2 M$\Omega$, TOC < 5 ppb, PURELAB$^{®}$ Ultra Analytic, ELGA (Veolia Water Technologies))" to (P5L16): "(18.2 M$\Omega$, TOC < 5 $\mu$g L-1, PURELAB$^{®}$ Ultra Analytic, ELGA (Veolia Water Technologies))."

Please also note the supplement to this comment:
https://www.atmos-meas-tech-discuss.net/amt-2017-363/amt-2017-363-AC2-supplement.pdf

---

## Author Comment (AC4) · 16 Mar 2018

Please find below the author's response to the interactive discussion comments from the anonymous referee.

Referee comments: RC

Author's response given below individual referee comments

RC- The paper "Field and laboratory evaluation of a high time resolution x-ray fluo-

[Figure]

rescence instrument for determining the elemental composition of ambient aerosols" by Tremper et al. is a comprehensive study aiming at providing laboratory and in-field information on the performance of the X-ACT 625 instrument. The paper is well structured, data presentation is adequate and well commented. X-ACT is an innovative instrument and improving its characterization is important for the scientific community. Thus, in my opinion the paper is of interest for publication in AMT and publication can occur after few revisions are performed.

Author's response: We would like to thank the anonymous referee for the positive feedback and the detailed suggestions to our manuscript. Please find the answers to the individual comments below. The original comments are written in black and the author's reply and changes to the manuscript are coloured in blue/green, respectively.

RC - Major concerns: RC - P10L18: why is Cd not mentioned? Cd is the element providing the strongest differences in all cases, but its discussion is completely missed in the text. Please add comments about it, or give explanation why it should be rejected. In this case, please remove it throughout the manuscript.

Author's response: Cd was mostly below the detection limit and thus did not give meaningful results in the Deming regression. To make this clearer the text section 3.2.1 was changed from "For the remaining elements (Ni after HF/HClO4 digestion and Cu and Se after HNO3/H2O2 digestion) the concentrations measured by the XACT were significantly lower than those measured by the ICP-MS. Cr and V were not reported for HF/HClO4 due to contamination of the HClO4 used in the digestion. The remaining elements were mostly below the limit of detection and thus did not produce meaningful regression results."

to (P10L28 onwards) "For the elements Ni (after HF/HClO4 digestion) Cu and Se (after HNO3/H2O2 digestion) the concentrations measured by the XACT were significantly lower than those measured by the ICP-MS. Cr and V were not reported for HF/HClO4 due to contamination of the HClO4 used in the digestion. In case of Cd and Ce a large

number of concentrations were below the LOD, and thus the elements were excluded from further comparison." For clarity and completeness in the description of the method these elements have been left in the rest of the manuscript.

RC - P11L30-35: Parallel sampling of PM10 and PM2.5 is needed to determine PM2.5/PM10 element ratios. Sampling different aerosol size fractions in different seasons and using ratios to separate fine and coarse contributions by elements is misleading. Please remove. X-ACT is an on-line device providing elemental composition of atmospheric aerosol. Other instruments (e.g. streaker sampler, rotating drum impactors) can provide high time-resolved measurements by off-line analyses performed at accelerator facilities (e.g. by Particle-Induced X-Ray Fluorescence or Synchrotron Radiation XRF). I think a comparison with such measurements should be mentioned as a perspective.

Author's response: In the study mentioned (Visser et al., 2015), different size fractions were sampled in parallel using a rotating drum impactor. This, however, was not clear in the text and we have changed "Measured chemical composition of different size fractions at Marylebone Rd during winter and summer campaigns during 2012 and the percentage of the element in the PM10-2.5 fraction can be used to highlight how these elements are distributed between the fine and coarse particle sizes: S 35 %, K 57 %, Ca 72 %, and Cl 73%..."

to (P12L2) "The chemical composition of different size fractions was sampled using a rotating drum impactor (RDI) and analysed with synchrotron radiation-induced X-ray fluorescence spectrometry (SR-XRF) during a winter campaign at Marylebone Road in 2012 (Visser et al., 2015b) and the percentage of the element in the PM10-2.5 fraction can be used to highlight how these elements are distributed between the fine and coarse particle sizes: S 35 %, K 57 %, Ca 72 %, and Cl 73%."

RC - Minor concerns:

RC - P2L2: please evidence that modelling approaches (and not only measurements

of aerosol chemical composition) are needed to gain information on aerosol sources

Author's response: To evidence that modelling approaches are needed the authors have changed the introductory sentence "Measuring the chemical composition of airborne particulate matter (PM) can provide valuable information on the concentration of regulated toxic metals and their sources and assist in the identification and validation of abatement techniques."

To (P2L1onwards) "Measuring the chemical composition of airborne particulate matter (PM) can provide valuable information on the concentration of regulated toxic metals, support modelling approaches for sources detection and assist in the identification and validation of abatement techniques."

RC - P3L2: please add ion chromatography for inorganic ions (as it is cited in the following and applied in the paper)

Author's response: The authors have included inorganic ions in the following sentence: "These filters are collected over a period of time, usually 24 hours to a week, and then analysed for different components such as metals (Brown et al., 2008), polyaromatic hydrocarbons (Pandey et al., 2011), elemental and organic carbon (Chu, 2004)."

It now reads (P3L3-5): "These filters are collected over a period of time, usually 24 hours to a week, and then analysed for different components such as metals (Brown et al., 2008), polyaromatic hydrocarbons (Pandey et al., 2011), elemental and organic carbon (Chu, 2004) and inorganic ions (Beccaceci et al., 2015)."

RC - P3L23: "sample a narrower range of components". Please change "sample" with "measure" (instruments sample what is in air but are not always able to quantify)

Author's response: Changed "sample" to "measure" in the following sentence (P3L26): "Furthermore, the high time resolution instruments tend to measure a narrower range of components with a higher Limit of Detection (LOD) than equivalent laboratory based methods, generally because less material is collected on each sample."

RC - P5L3: wrong formula for Ammonium sulphate (cfr. P5L13 where it is correct)

Author's response: Corrected the formula for Ammonium sulphate (P5L14): "Ammonium sulphate ((NH4)2SO4, ACS reagent grade, Sigma-Aldrich) . . ."

RC - P6L6: "where source contributions may be assumed based on one of these measurement techniques". I guess the authors refer to receptor modelling approaches for source apportionment. Please change into "where source contributions may be estimated by receptor modelling using measurements of chemical components as input" (the measurement techniques do not provide source contributions, but quantify chemical components)

Author's response: The authors amended the sentence incorporating the referee's suggestion. The sentence "Although the measurands are not directly comparable, they provide useful information for studies where source contributions may be assumed based on one of these measurement techniques." now reads (P6L7onwards): "Although the measurands are not directly comparable, they provide useful information for studies where source contributions may be estimated by receptor modelling using measurements of chemical components based on one of these measurement techniques."

RC - P7L3-5: obscure. Please add some explanation

Author's response: For clarification the authors have changed the following sentence: "The measurements were quality assured against measurements of SMPS (for volume to ensure the collection efficiency is suitable) and PM2.5 mass when combined with Aethalometer measurements as described by Crenn et al. (2015)."

It now reads (P7L5) "The ACSM measurements were combined with Aethalometer measurements and compared to PM2.5 mass measured using the TEOM FDMS or PM1 mass estimated using SMPS measurements as described by Crenn et al. (2015)."

RC - P8L7: 3 or sqrt(3)?

Author's response: To clarify, the format of the formula was changed from on from" For the XACT measurements, the combined uncertainty included contributions of $3/\sqrt{3}$% from flow (CEN, 2014) . . ." to (P8L14): "For the XACT measurements, the combined uncertainty included contributions of $3/(\sqrt{3})$ % from flow (CEN, 2014) . . ."

RC - P8L12&15: what is k?

Author's response: The following text has been changed: " For the ACSM, the sulphate measurement uncertainty was estimated as 14 % (k = 2) for sulphate at a 30-min resolution by Crenn et al. (2015) and the LOD was determined using HEPA field blank measurements as 34.9 ng m-3. For the URG, the chloride and sulphate LODs were reported by the manufacturer as 100 ng m-3 and verified by Beccaceci et al. (2015). The uncertainty of the species measured by ion chromatography was estimated at 4.5 % (k = 2) by Yardley et al. (2007) and combined with the additional 97 % extraction efficiency of a particle-to-liquid sampler system estimated by Orsini et al. (2003)." to explain k (coverage factor).

It now reads (P8L18 onwards): "For the ACSM, the sulphate measurement uncertainty was estimated as 14 % (coverage factor k = 2) for sulphate at a 30-min resolution by Crenn et al. (2015) and the LOD was determined using HEPA field blank measurements as 34.9 ng m-3. For the URG, the chloride and sulphate LODs were reported by the manufacturer as 100 ng m-3 and verified by Beccaceci et al. (2015). The uncertainty of the species measured by ion chromatography was estimated at 4.5 % (coverage factor k = 2) by Yardley et al. (2007) and combined with the additional 97 % extraction efficiency of a particle-to-liquid sampler system estimated by Orsini et al. (2003)."

RC - P8L27: Are the authors referring to statistical significance?

Author's response: Yes, the authors are referring to statistical significance as the results of the Deming regression are given at a 95% confidence interval. To clarify this in the text, the CI was added to the sentence "Slopes are not significantly different from the

1:1 line for all comparisons."

and it now reads (P8L36): "Slopes are not significantly different from the 1:1 line for all comparisons (95% confidence interval)."

RC - P9L1: "dominated by fireworks activity (Oct-Dec 2014)". Do the authors mean that fireworks were the main source impacting the area in those 3 months? If not, please rephrase.

Author's response: During this sampling period there were a number of events and thus peak concentrations were dominated by fireworks rather than fireworks being the main source overall. To clarify this, the sentence was changed from "The sampling at Marylebone Road was carried out using a PM2.5 inlet during a period that was dominated by fireworks activity..."

to (P9L7) "The sampling at Marylebone Road was carried out using a PM2.5 inlet during a period when peak concentrations were dominated by fireworks activity..."

RC - P9L7-19: First of all, suitable references to fireworks tracers in aerosol are missing. Secondly, high time resolved measurements of elemental composition during fireworks has already been presented in the literature. In one case, they were also exploited for source apportionment by receptor models (Vecchi et al., 2008. DOI: 10.1016/j.atmosenv.2007.10.047)

Author's response: The authors have now included suitable references and have also highlighted the fact that high time resolved measurements of elemental composition during fireworks has been use in source apportionment before. The section has been changed as follows (with the added references and text changes underlined) and the reference list has been updated (P9L7 onwards): "The sampling at Marylebone Road was carried out using a PM2.5 inlet during a period when peak concentrations were dominated by fireworks activity (Oct-Dec 2014). The mean concentrations across all elements measured during this campaign ranged from 0.177 ng m-3 to 600 ng m-3

and elements typically used in fireworks such as Ba, Sr, K and Ti (Godri et al., 2010; Moreno et al., 2007; Vecchi et al., 2008) had high maximum concentrations. Traffic emissions further influenced the metal concentrations at Marylebone Road. Overall the order of the elements in terms of mean concentration was: S > Fe > Cl > K > Si > Ca > Zn > Cu > Ba > Pb > Mn > Ti > Cd > Sr > As > Cr > Ce > V > Ni > Mo > Pt > Se. This dataset helps highlight that high time resolution data has the advantage of giving much more detailed information on high pollution events, which can be used e.g. in source apportionment (Vecchi et al., 2008) and for health studies (Godri et al., 2010; Hamad et al., 2016). Figure 3 shows the daily filter and hourly XACT measurements of K and Ba during a period of increased bonfire and fireworks activity due to Diwali (Hindu festival of light) and Guy Fawkes celebrations. The daily filter measurements show that the highest concentrations of K, which is used as an oxidiser in fireworks (Moreno et al., 2007) but also a tracer for biomass burning, were measured on the 5th and 6th November 2014, followed by slightly lower concentrations on the 7th and 8th of November. On the other hand Ba, which is used in green fireworks (Moreno et al., 2007), displays similarly high concentrations on all four days. Looking at the K concentration in a higher time resolution as measured by the XACT, it is evident that peak concentrations were comparable on the nights of the 5th, 7th and 8th of November (data is missing for the 6th of November due to instrument failure) but the high concentrations did not last as long on the 7th and 8th of November. The highest Ba concentration on the other hand was measured on the 8th of November with lower concentrations on the 5th and 7th. This difference in contribution might point to different fireworks being used."

RC - P9L22&34: "mean concentrations". Please change into "mean elemental concentrations"

Author's response: Thanks to this comment and a further comment by anonymous referee 4, the authors realised that it was not clear which concentration/elements they were referring to. Thus the following sentences were changed: "The mean concentrations measured in this campaign ranged from 0.24 ng m-3 to 5,200 ng m-3."

Now reads (P9L30): "Overall, the mean elemental concentrations measured in this campaign ranged from 0.24 ng m-3 to 5,200 ng m-3." "The influence of the local industry in Tinsley, Sheffield was reflected by high concentrations of metals like Ni and Cr, with mean concentrations more than 30 times that found in the Marylebone Road campaign with mean concentrations ranged from 0.186 ng m-3 to 1,370 ng m-3." Now reads (P10L3): "The influence of the local industry in Tinsley, Sheffield was reflected by high concentrations of metals like Ni and Cr, with mean concentrations more than 30 times that found in the Marylebone Road campaign. The mean elemental concentrations overall ranged from 0.186 ng m-3 to 1,370 ng m-3."

RC - P10L1&P11L2: how was non-sea sulphate calculated? Please describe or add suitable references. Furthermore, provide references for this choice in the comparison.

Author's response: The method for calculating non-sea salt sulphate has been included in section 3.2.2. The following sentence was expanded: "The hourly values of S and Cl measured with the XACT were used to calculate hourly non-sea salt sulphate (SO4), which was compared to the hourly sulphate (predominantly ammonium sulphate) which is non-refractory measured by the ACSM (Chang et al., 2011)."

It now reads (P11L10 onwards): "The hourly values of S and Cl measured with the XACT were used to calculate hourly non-sea salt sulphate (SO4) based on their relative abundance in sea water (Millero et al. 2008). It should be noted that Cl is used in the absence of the preferred Na and Cl concentration measured could be partially depleted by reaction between NaCl and nitric acid (HNO3). The hourly non-sea salt sulphate was compared to the hourly sulphate (predominantly ammonium sulphate) which is non-refractory measured by the ACSM (Chang et al., 2011)." The reference Millero et al. 2008 was added to the reference list: Millero, F. J., R. Feistel, D. G. Wright and T. J. McDougall (2008). "The composition of Standard Seawater and the definition of the Reference-Composition Salinity Scale." Deep Sea Research Part I: Oceanographic

Research Papers 55(1): 50-72.

RC - P10L28: "filter artefacts"? What do the authors refer to? Sampling artefacts or something else? Please, clarify.

Author's response: Following comments by referee 4 the authors have removed the following "…positive and negative filter artefacts could also influence the concentrations when sampling onto filters…" and revised the section in question. The authors agree that the elements compared between XACT and filter based measurements (analysis with ICP-MS) are not influenced by sampling artefacts per se but there are differences between the sampling methodologies which could result in differences in concentration and these are now reflected in this section.

The section now reads (P10L33onwards): "There are a variety of possible reasons for the differences observed between the methods. In the case of the filter analysis, the blank filters were found to be variable and thus subtracted values may result in an under- or overestimation of the true concentration; the digestion recovery rates were not taken into account; many concentrations were close to the detection limit for the elements As in all campaigns and Ni during the Marylebone Road campaign. These stated reasons might influence the two digestions methods to different extents. Unfortunately, there was no opportunity to undertake both digestions on the same samples. To provide some insight into how the two digestion methods compared, the XACT measurements were grouped into concentration appropriate bins and the associated ICP-MS measurements from each digestion method were averaged and compared. These are shown in S6 (Deming regression of ICP-MS using different digestion methods). For the XACT, the standards used in calibrations were much higher than ambient concentrations and the calibration matrix differed from sample matrix (Indresand et al., 2013). Despite every effort being made to co-locate the sample inlets in all field trials, slight differences in inlet location, especially when close to the road, could not be avoided. This and different temperatures of the sample inlets may also contribute to differences observed in concentrations. Nevertheless, the results of the XACT comparison with
ICP-MS in this study are comparable to those reported in other studies (Furger et al., 2017)."

RC - Typos

RC - P2L31: "implementing" instead of "implement"

Author's response: The authors believe that "... helps implement policies..." is the correct expression.

RC - P7L1 vs P7L2: Middlebook or Middlebrook?

Author's response: The spelling of "Middlebrook" has been corrected on P7L4: "The collection efficiency was calculated using the Middlebrook parameterisation (Middlebrook et al., 2012)..."

RC - P10L30: change "extends" into "extents"

Author's response: The sentence on P10L38 was corrected from "These stated reasons might influence the two digestions methods to different extends." to "These stated reasons might influence the two digestions methods to different extents."

RC - P11L7: ".." Change into "."

Author's response: Removed the duplicated "." "...resulted in higher results than off-line filter measurements."

Please also note the supplement to this comment:
https://www.atmos-meas-tech-discuss.net/amt-2017-363/amt-2017-363-AC4-supplement.pdf

---

## Author Comment (AC5) · 16 Mar 2018

Please find below the author's response to the interactive discussion comments from the anonymous referee.

Referee comments: RC

Author's response given below individual referee comments

RC - Overall the manuscript presents relevant results in a comprehensive and wellorganized manner. So the publication in AMT is recommended. Still some conclusions need to be revised or softened, or better supported. Please see the comments below.

Author's response: We would like to thank the anonymous referee for the positive feedback and the suggestions to our manuscript. Please find the answers to the individual comments below. The original comments are written in black and the author's reply and changes to the manuscript are coloured in blue/green, respectively.

RC - Abstract. State which elements are compared with ICP and which with 'other high time resolution measurements', because the differences in slopes (median 1.07 vs 1.41-4.6) may be due to the different elements being assessed rather than to differences between techniques?

Author's response: The authors have added the elements to the relevant sections in the abstract: "The XRF technique agreed well with the ICP-MS measurements of daily filter samples in all cases with a median R2 of 0.93 and a median slope of 1.07. Differences were likely due to recovery rates from the sample digestion as well as filter sampling artefacts and matrix effects in the XRF technique. The XRF technique also agreed well with the other high time resolution measurements but showed a significant positive bias (slopes between 1.41 and 4.6), probably due to differences in the size selection methodology, volatility and water solubility of the PM in aerosol mass spectrometry and ion chromatography, respectively."

It now reads (P2L15 onwards, please note this section changed further due to other comments): "The XRF technique agreed well with the ICP-MS measurements of daily filter samples in all cases with a median R2 of 0.93 and a median slope of 1.07 for the elements As, Ba, Ca, Cr, Cu, Fe, K, Mn, Ni, Pb, Se, Sr, Ti, V and Zn. Differences in the results were attributed to a combination of inlet location and sampling temperature, variable blank levels in filter paper and recovery rates from acid digestion. The XRF technique also agreed well with the other high time resolution measurements but showed a clear positive difference (slopes between 1.41 and 4.6), probably due to differences in the size selection methodology, volatility and water solubility of the PM in aerosol mass spectrometry (SO4) and ion chromatography (Ca, Cl, K, SO4), respectively."

RC - Page 3. Line 4. The positive and negative sampling artefacts are true for some species but not for others, e.g. metals concentrations determined on filter samples by digestion+ICP do not suffer from sampling artefacts.

Author's response: The authors agree with the above statements and have thus amended the following sentence: "This approach is time consuming, labour intensive and prone to positive and negative sampling artefacts (Chow et al., 2015)."

It now reads (P3L6): "This approach is time consuming, labour intensive and prone to positive and negative sampling artefacts for some components (Chow et al., 2015)."

RC - Page 3. Lines 34-36. Furger et al. (2017) used both ICP-OES and ICP-MS, not only ICP-MS for the list of elements reported in this manuscript.

Author's response: The authors have included ICP-OES and gold amalgamation atomic absorption spectrometry in the following sentence: "Furger et al. (2017) tested the XACT during a summer campaign in Switzerland in 2015 and compared the XACT data with measurements made using ICP-MS on filters sampled for 24 hours (both PM10) as well as ACSM measurements (PM1)."

It now reads (P3L40 onwards): "Furger et al. (2017) tested the XACT during a summer campaign in Switzerland in 2015 and compared the XACT data with measurements made using ICP-OES (inductively coupled plasma optical emission spectrometry) , ICP-MS and gold amalgamation atomic absorption spectrometry on filters sampled for 24 hours (both PM10) as well as ACSM measurements (PM1)."

RC - Page 6. Lines 39-40 and page 7, line 1. Wasn't the RIE for ammonium calculated from the calibration with ammonium nitrate? And hence only the RIE for sulphate calculated from calibration with ammonium sulphate? Please correct if necessary.

[Figure]

Author's response: The authors have amended the following sentences: "The ionisation efficiency was calculated using a mono-disperse supply of ammonium nitrate aerosols that were size selected through a differential mobility analyser and counted using a condensation particle counter (CPC). The relative ionisation efficiencies of sulphate and ammonium were calculated from separate calibrations using a mono-disperse supply of ammonium sulphate aerosols."

It now reads (P6L39 onwards): "The ionisation efficiency of nitrate and the relative ionisation efficiencies of ammonium and sulphate were calculated using a mono-disperse supply of ammonium nitrate and ammonium sulphate aerosols. These were size selected through a differential mobility analyser and counted using a condensation particle counter (CPC) as described by Crenn et al. (2015)."

RC - Page 9, lines 22-23. Please modify the sentence. As written now it seems you are still taking about the Ni, and according to Table 4, Ni concentration is reported to be 20 and min and max 0.24 and 320. Hence, from Table 4, one can see that 0.24-5200 is the range of mean concentrations for all the species analysed, but from the text is not clear at all.

Author's response: Thanks to this comment and a further comment by anonymous referee 3, the authors realised that it was not clear which concentration/elements they were referring to. Thus the following sentences were changed: - "The mean concentrations measured in this campaign ranged from 0.24 ng m-3 to 5,200 ng m-3."

Now reads (P9L30): "Overall, the mean elemental concentrations measured in this campaign ranged from 0.24 ng m-3 to 5,200 ng m-3."

- "The influence of the local industry in Tinsley, Sheffield was reflected by high concentrations of metals like Ni and Cr, with mean concentrations more than 30 times that found in the Marylebone Road campaign with mean concentrations ranged from 0.186 ng m-3 to 1,370 ng m-3."

Now reads (P10L3): "The influence of the local industry in Tinsley, Sheffield was reflected by high concentrations of metals like Ni and Cr, with mean concentrations more than 30 times that found in the Marylebone Road campaign. The mean elemental concentrations overall ranged from 0.186 ng m-3 to 1,370 ng m-3."

RC - Page 9, line 25. The reason for high Cl concentration is not only the PM10 head, as at Tinsley the head was also PM10 but Cl is not so high. I guess the proximity to ocean played a role here.

Author's response: The proximity of the ocean will certainly have played a role, and thus we included the site characteristic in our example in the following sentence: "The concentrations and dominant elements will be influenced by the site characteristics as well as the size range sampled; e.g. Cl from sea salt is predominantly found in the coarse fraction and thus much higher at Pontardawe as sampling was carried out using a PM10 head."

It now reads (P9L31 onwards): "The concentrations and dominant elements will be influenced by the site characteristics as well as the size range sampled; e.g. Cl from sea salt is predominantly found in the coarse fraction and thus much higher at Pontardawe as the sample site is closer to the sea and sampling was carried out using a PM10 head."

RC - Page 10, line 2. With hourly concentration ranging. . .? Is it hourly? Or?

Author's response: The concentrations used in the comparison with the ACSM and URG are indeed hourly, which was not clear in the text and has been changed: "The mean concentration of non-sea salt sulphate (XACT) and non-refractory sulphate (ACSM) during the fireworks campaign at Marylebone Road was 2,600 ng m-3 and 2,000 ng m-3, respectively, with concentration ranging from 240 ng m-3 to 10,500 ng m-3 SO4 (non-sea salt) and 58 ng m-3 to 8,300 ng m-3 for non-refractory SO4. The comparison of the XACT with the URG was carried out in PM10 during winter 2014/2015. The concentration of water soluble anions and cation ranged from 154 ng

m-3 (K) to 1,790 ng m-3 (Cl) compared to 145 ng m-3 (K) to 2,700 ng m-3 (Cl) in total element concentrations. "

It now reads (P10L8 onwards) "The mean hourly concentration of non-sea salt sulphate (XACT) and non-refractory sulphate (ACSM) during the fireworks campaign at Marylebone Road was 2,600 ng m-3 and 2,000 ng m-3, respectively, with hourly concentration ranging from 240 ng m-3 to 10,500 ng m-3 SO4 (non-sea salt) and 58 ng m-3 to 8,300 ng m-3 for non-refractory SO4. The comparison of the XACT with the URG was carried out in PM10 during winter 2014/2015. The hourly concentration of water soluble anions and cation ranged from 154 ng m-3 (K) to 1,790 ng m-3 (Cl) compared to 145 ng m-3 (K) to 2,700 ng m-3 (Cl) in total element concentrations. "

To further clarify this point, we have changed the caption of Table 6 : "Table 2: Overview of Marylebone Road, London SO4 measurements in PM2.5 by XACT (SO4* calculated as non-sea salt SO4 using S and Cl measurements) and ACSM (ng m-3); and SO4, K, Cl, Ca measurements in PM10 by XACT (SO4** calculated as predicted SO4 using S measurements) and URG (ng m-3)"

It now reads (P22L3 onwards) "Table 3: Overview of Marylebone Road, London hourly SO4 measurements in PM2.5 by XACT (SO4* calculated as non-sea salt SO4 using S and Cl measurements) and ACSM (ng m-3); and hourly SO4, K, Cl, Ca measurements in PM10 by XACT (SO4** calculated as predicted SO4 using S measurements) and URG (ng m-3)"

RC - Page 10, line 8. They are not ICP-MS digestion methods, they are digestion methods. The ICP-MS is used afterwards. Please re-write.

Author's response: The text has been changed from "The filter comparison results were split by the two ICP-MS digestion methods..." to (10L16) "The filter comparison results were split by the two digestion methods..."

RC - It is not so clear that the differences XACT vs ICP can be attributed to the recovery

rates in the digestion processes prior to ICP analysis. Please amend through the manuscript this explanation (especially conclusions, page 12 lines 15 and following, line 28). One needs to asses this statement based on the individual elements. If that statement was true, the elements with the lowest recovery rates would have highest slopes, but this is not the case e.g. recovery for Ni (HF/HClO4) is 87

Author's response: Please see combined response with next point as the comments overlapped.

RC - In the conclusions, again, please make sure you don not attribute the difference between XACT and ICP to filter artefacts when you discuss elements that do not suffer from filter artefacts (page 12, line 28, among others).

Author's response: The authors accept that it was not clear from the manuscript that the differences of the XACT and filter based method followed by analysis with ICP-MS are caused by a multitude of reasons. Some of the potential reasons had been omitted and are now included in the current revision Also the authors agree that the elements compared do not suffer from what were loosely described as filter artefacts and thus this was amended accordingly. Please see the changes below.

In the abstract the following sentence was changed: "Differences were likely due to recovery rates from the sample digestion as well as filter sampling artefacts and matrix effects in the XRF technique."

It now reads (P2L17 onwards): "Differences in the results were attributed to a combination of inlet location and sampling temperature, variable blank levels in filter paper and recovery rates from acid digestion."

In section 3.2.1 the authors have removed the following "...positive and negative filter artefacts could also influence the concentrations when sampling onto filters..." and revised the section in question.

The section now reads (P10L33onwards): "There are a variety of possible reasons

for the differences observed between the methods. In the case of the filter analysis, the blank filters were found to be variable and thus subtracted values may result in an under- or overestimation of the true concentration; the digestion recovery rates were not taken into account; many concentrations were close to the detection limit for the elements As in all campaigns and Ni during the Marylebone Road campaign. These stated reasons might influence the two digestions methods to different extents. Unfortunately, there was no opportunity to undertake both digestions on the same samples. To provide some insight into how the two digestion methods compared, the XACT measurements were grouped into concentration appropriate bins and the associated ICP-MS measurements from each digestion method were averaged and compared. These are shown in S6 (Deming regression of ICP-MS using different digestion methods). For the XACT, the standards used in calibrations were much higher than ambient concentrations and the calibration matrix differed from sample matrix (Indresand et al., 2013). Despite every effort being made to co-locate the sample inlets in all field trials, slight differences in inlet location, especially when close to the road, could not be avoided. This and different temperatures of the sample inlets may also contribute to differences observed in concentrations. Nevertheless, the results of the XACT comparison with ICP-MS in this study are comparable to those reported in other studies (Furger et al., 2017)."

In the conclusions the following sentences were changed: - "This was attributed to recovery rates from acid digestion and filter sampling."

It now reads (P12L30 onwards): "Differences in the individual results were element specific but generally attributable to a combination of variable filter blank levels, recovery rates from acid digestion, instrument calibration, sampling temperature and small differences in inlet location."

- "This suggests that the XACT accurately measures elemental ambient aerosol composition and that the positive bias, when compared to the ICP-MS measurements identified in the field experiments, was more likely due to filter artefacts and recovery rates

following acid digestion."

It now reads (P13L6 onwards): "This suggests that the XACT accurately measures elemental ambient aerosol composition and that the positive bias, when compared to the ICP-MS measurements identified in the field experiments, was not due to the XACT calibration but more likely due to the remaining reasons listed above."

RC - Conclusions, page 12, lines 17-18. If the sampling size was different and it is true that the size range 1-2-5 um has so much sulphate, then the 1.68 is not a bias. The 1.68 is not a bias but would have actual meaning. Please re-phrase.

Author's response: The authors agree with the referee that the difference in slope has actual meaning. To make this clearer they have changes the wording in the following section: "When compared to the alternative aerosol mass spectrometry and ion chromatography based high time resolution techniques, the XACT showed good temporal agreement but with a significant positive bias (median 1.68) compared to the ICP-MS; this was likely due to the differences in the size selection methodology employed by the different techniques as well as particle volatility and water solubility. However, these differences in solubility and volatility could be utilised to provide information about different sources and their contributions; such as the difference between refractory sodium chloride and non-refractory ammonium chloride."

It now reads (P12L32 onwards): "When compared to the alternative aerosol mass spectrometry and ion chromatography based high time resolution techniques, the XACT showed good temporal agreement but with a clear positive difference (median 1.68) compared to the ICP-MS; this was likely due to the differences in the size selection methodology employed by the different techniques as well as particle volatility and water solubility. However, these differences (size, solubility and volatility) could be utilised to provide information about different sources and their contributions; such as the difference between refractory sodium chloride and non-refractory ammonium chloride." The introduction was amended accordingly.

RC - Technical corrections:

RC - Page 7, line 8. Shouldn't it say ". . .described in Beccaceci et al. (2015)"?

Author's response: Changed the referencing on P7L10 from ". . .described by (Beccaceci et al., 2015) to: "The URG-900B Ambient Ion Monitor continuously measured water-soluble anion and cation concentrations (Cl-, SO42-, NO3-, Na+, NH4+, K+, Mg2+, and Ca2+) in PM10 and is described in Beccaceci et al. (2015)."

RC - Page 8, line 11. Remove 1 point after "Table 3".

Author's response: Removed the duplicated "." ". . .; these are shown in Table 3. For the ACSM. . ."

RC - Page 8, line 13. A space is missing before "For".

Author's response: Included a space before "For" on P8L20: ". . .and the LOD was determined using HEPA field blank measurements as 34.9 ng m-3. For the URG. . ."

Please also note the supplement to this comment:
https://www.atmos-meas-tech-discuss.net/amt-2017-363/amt-2017-363-AC5-supplement.pdf

---

## Author Comment (AC6) · 16 Mar 2018

Please find below the author's response to the interactive discussion comments from the anonymous referee. Referee comments: RC

Author's response given below individual referee comments

RC - The new version of the article has been greatly improved. Some modification have still to be done before publication:

Author's response: We would like to thank the anonymous referee for the feedback and the suggestions to our manuscript. Please find the answers to the individual comments below.

RC - 1) In section 3.1 the authors should explicitly cite the fact that in XRF there are neither self-absorption problems for the medium-high Z elements nor matrix effects. There can be self-absorption effects only for low Z elements (the only one, which can be affected in their comparison, is Si) even with samples with high loading unless the deposit on a very small area. Those effects depend mostly on the absorption within the single particle therefore they are present also in the samples prepared by the authors. The use of self-made standard can be useful, but I do not see any problem in the use of commercial standards as it is done in many laboratories which routinely use XRF for aerosol analysis.

Author's response: The authors agree that self-absorption would not be a significant problem even at the thickness of deposit and composition encountered in this experiment. Please also see response to referee 1. Nevertheless the authors have changed the sentence in section 3.1: "All calibrations resulted in a linear relationship between the mass calculated using TEOM mass concentrations and measured by the XACT for the standard range used."

It now reads (P8L32 onwards): "All calibrations resulted in a linear relationship between the mass calculated using TEOM mass concentrations and measured by the XACT for the standard range used. Sample self absorption effects were calculated to be <1% for the maximum concentration of S (the lightest element used) and therefore insignificant in the use of this instrument."

A common criticism of the commercial standards is that they are 1) not at the concentration range expected from ambient air sampling and 2) not on the same filter matrix as those typically used in ambient air sampling. The development of new calibration techniques at a wider range of concentrations and using different compounds provides

a method of validating the current quality assurance techniques. This is evidenced by the comparison between the XACT and TEOM that is linear down to concentration levels well below those found on commercial XRF standards.

RC - 2) Again what is reported at the end of section 3.2.1 page 11 lines 12-13 is not correct (same comment as above)

Author's response: In section 3.2.1 the authors address the common criticism of XRF standards and believe it would be an oversight not to do so. See also answer to comment above.

RC - 3) Section 3.2.2: the use of Cl to calculate non sea-salt sulphate can give a strong overestimation of that component due to the possible volatilization of Cl in case of aged sea-salt as reported in many works regarding also the sites analyzed by the authors. Normally Na is used. The authors must make a comment about this

Author's response: The method used for calculating non-sea salt sulphate has been included in section 3.2.2. and reference is made to the possible depletion of Cl. The following sentence was expanded: "The hourly values of S and Cl measured with the XACT were used to calculate hourly non-sea salt sulphate (SO4), which was compared to the hourly sulphate (predominantly ammonium sulphate) which is non-refractory measured by the ACSM (Chang et al., 2011)."

It now reads (P11L10 onwards): "The hourly values of S and Cl measured with the XACT were used to calculate hourly non-sea salt sulphate (SO4) based on their relative abundance in sea water (Millero et al. 2008). It should be noted that Cl is used in the absence of the preferred Na and Cl concentration measured could be partially depleted by reaction between NaCl and nitric acid (HNO3). The hourly non-sea salt sulphate was compared to the hourly sulphate (predominantly ammonium sulphate) which is non-refractory measured by the ACSM (Chang et al., 2011)." The reference Millero et al. 2008 was added to the reference list: Millero, F. J., R. Feistel, D. G. Wright and T. J. McDougall (2008). "The composition of Standard Seawater and the definition of

the Reference-Composition Salinity Scale." Deep Sea Research Part I: Oceanographic Research Papers 55(1): 50-72.

RC - 4) All the information reported by the authors are interesting and better explained in this new version. However, in my opinion, it cannot be neglected that the best way to assess the performances of the XACT spectrometer would have been to use a standard aerosol sampler (like one of those used by the authors), the proper collection filters (e.g. Teflon or polycarbonate filters) and XRF analysis of the collected filters. The authors should make an explicit comment about this.

Author's response: The authors believe that an instrument field evaluation needs to include commonly used reference methods, such as the European reference method EN14902 and other studies, such as the verification test carried out by the US-EPA (US-EPA, 2012) have taken a similar approach. As reference methods are used for regulatory purposes this provides the context in which element concentrations and their changes are viewed; other commonly used techniques were included in the field analysis, which was not claimed to be exhaustive.

The US-EPA study was added to the introduction as explained in comment to Referee 2 and the reference was added to the reference list: "US-EPA: Environmental Technology Verification Report. Cooper Environmental Services LLC Xact 625 Particulate Metals Monitor, Report no. EPA/600/R-12/680. Agency, U. S. E. P. A. (Ed.), U.S. Environmental Protection Agency, Cincinnati, OH 45268, 2012."

However, the filter analysis technique using the XACT and piloted in this study would allow a direct comparison of the XACT and other XRF systems as mentioned in the conclusions (13L15).

Please also note the supplement to this comment:
https://www.atmos-meas-tech-discuss.net/amt-2017-363/amt-2017-363-AC6-supplement.pdf

---

## Author Response (AR2)

**Associate Editor Decision: Publish subject to minor revisions (review by editor)**
(04 Apr 2018) by Willy Maenhaut

Comments to the Author:
The authors have reasonably addressed the comments of the five anonymous referees and they have modified their manuscript accordingly. However, the comments below should be taken into consideration for the main text and the Supplement before the manuscript can be published in AMT.
**Author's response:** The authors would like to thank the editor for the positive response and for carefully looking through the manuscript and the suggestions made. Please find below the detailed response to the comments.

Main text:
Page 2, line 2: replace "sources detection" by "source detection".
**Author's response:** Corrected "sources detection" to "source detection"

Page 2, line 7: Pd should be removed from the list of elements as it was not measured in the PM; as indicated on page 5, lines 10-11, a Pd rod was used for the internal standard measurement.
**Author's response:** Removed Pd from the element list as the editor correctly pointed out that it is not measured in PM but used as internal standard.

Page 3, line 38: it is unclear for which technique Cu was close to the detection limit; this should be specified.
**Author's response:** Cu was close to the limit of detection of" the ICP-MS and close to the quantitation limit of the XACT. To make this clearer in the manuscript, the text has been changed from:
"In a verification test carried out by the US-EPA (2012) measurements of Ca, Cu, Mn, Pb, Se and Zn by the XACT were compared to filter based measurements (filters analysed using ICP-MS). This verification test showed that the daily average Xact 625 results were highly correlated and in close quantitative agreement with ICP-MS analysis results for the six metals, except Cu, which was close to the detection limit in many cases."
It now reads:
"In a verification test carried out by the US-EPA (2012) measurements of Ca, Cu, Mn, Pb, Se and Zn by the XACT were compared to filter based measurements (filters analysed using ICP-MS). This verification test showed that the daily average Xact 625 results were highly correlated and in close quantitative agreement with ICP-MS analysis results for the six metals, except Cu, which was close to the detection limit of the ICP-MS analysis and the quantitation limit of the Xact 625."

Page 3, lines 39-40: replace "filter based measurement" by "filters".
**Author's response:** Changed "filter based measurement" to "filters".

Page 5, line 8: replace "reported are" by "measured are".
**Author's response:** Changed "reported are" to "measured are".

Page 6, line 37: replace "to100" by "to 100".
**Author's response:** Added space between "to" and "100".

Page 7, line 6: which PM inlet, if any, was used with the Aethalometer?
**Author's response:** Added size range of aethalometer measurements in brackets. It now reads:
"The ACSM measurements were combined with Aethalometer measurements ($PM_{2.5}$) and compared to $PM_{2.5}$ mass measured using the TEOM FDMS or $PM_1$ mass estimated using SMPS measurements as described by Crenn et al. (2015)."

Page 7, line 18, and page 12, line 19: I think that it should be "Zefluor" instead of "Zeflour"?
**Author's response:** The spelling of "Zefluor" was correct on P7/L18; the spelling was corrected from "Zeflour" to "Zefluor" on P12/L19.

Page 7, line 21: replace "25mm" by "25 mm".
**Author's response:** Changed "25mm" to "25 mm".

Page 7, line 31: insert a "." after "method".
**Author's response:** added a full stop to the end of the sentence on P7/L31.

Page 8, line 13: "CEN, 2014" is missing from the Reference list.
**Author's response:** The authors have added the missing reference to the reference list as follows:
"CEN: Ambient air - Standard gravimetric measurement method for the determination of the $PM_{10}$ or $PM_{2.5}$ mass concentration of suspended particulate matter (EN 12341:2014), European Committee for Standardization (CEN), Brussels. 2014."

Page 8, lines 28-30: the unit for the thin film standards in mass per cm2 while that for the elements in the PM is in mass per m3; how can the data then be compared?
**Author's response:** The comparison of the concentration used in the laboratory experiment and the thin film standard was done using the raw data (ng) as recorded by the XACT. This is not clear in the text and the sentence was changed from:
"Thus, the highest element concentrations in the standards used for comparison were between 9 (S) and 25 (Zn) times lower than the commercial thin film standards."
It now reads:
"The highest element concentrations in the standards used for comparison were between 9 (S) and 25 (Zn) times lower than the commercial thin film standards when compared as ng."

Page 9, line 5: rename "Table 3-6" by "Tables 3-6".
**Author's response:** Corrected "Table 3-6" to "Tables 3-6".

Page 10, line 12: it should be specified for which measurement location the EXACT and URG were compared.
**Author's response:** Added the sample location to P10/L12. The sentence now reads:
"The comparison of the XACT with the URG was carried out in $PM_{10}$ at Marylebone Road during winter 2014/2015. "

Page 10, line 13: replace "cation ranged" by "cations ranged".
**Author's response:** The authors replaced "cation ranged" with "cations ranged, and deleted the double space in two places in this sentence. The sentence now reads:
"The hourly concentration of water soluble anions and cations ranged from 154 ng m$^{-3}$ (K) to 1,790 ng m$^{-3}$ (Cl) compared to 145 ng m$^{-3}$ (K) to 2,700 ng m$^{-3}$ (Cl) in total element concentrations."

Page 10, line 19: rename "Table 3 -5" by "Tables 3-5".
**Author's response:** Corrected "Table 3 -5" to "Tables 3-5".

Page 10, line 31: it is unclear for which technique many Cd and Ce data were below the detection limit; this should be specified.
**Author's response:** Many of the hourly XACT measurements were below the LOD, making it difficult to calculate daily means. The authors have amended the sentence to read:
"In case of Cd and Ce a large number of hourly XACT concentrations were below the LOD, and thus the elements were excluded from further comparison."

Page 11, line 17: abbreviations and acronyms, here "CI" should be defined (written full-out) when first used.
**Author's response:** Included the acronym definition when first used. The sentence now reads:
"The correlation resulted in a slope of 1.41 (95% confidence interval (CI) 1.35-1.46) and an intercept of 53 (95% CI 13.4-93) ng m$^{-3}$."

Page 12, line 20: the measurement location ("at Marylebone Rd") should be added at the end of this line.
**Author's response:** Section 3.3 (P12L9-24) is not referring to the exposure at Marylebone Road but gives the results from the comparison of the field measurements of the XACT with filter measurement (analysed using the XACT in the laboratory) taken in Sheffield, Tinsley. To clarify this, the sample location has been added to the first sentence of the section:
"With a mean $R^2$ of 0.95 daily concentrations measured on the filter by the XACT compared well with the measurements made by the XACT when deployed in the field."
It now reads:
"With a mean $R^2$ of 0.95 daily concentrations measured on the filter by the XACT compared well with the measurements made by the XACT when deployed in the field in Tinsley, Sheffield."

Page 13, line 9: the measurement location ("at Tinsley, Sheffield") should be added at the end of this line.
**Author's response:** The sentence on P13L6-9 is referring to all field experiments and not just Tinsley, Sheffield. To clarify this, the sentence has been changed from:
"This suggests that the XACT accurately measures elemental ambient aerosol composition and that the positive

5   bias, when compared to the ICP-MS measurements identified in the field experiments, was not due to the XACT calibration but more likely due to the remaining reasons listed above."
It now reads:
"This suggests that the XACT accurately measures elemental ambient aerosol composition and that the positive bias, when compared to the ICP-MS measurements identified in the field experiments in all locations, was not

10   due to the XACT calibration but more likely due to the remaining reasons listed above."

Pages 13-15, Reference list: titles of journal articles should be in lower case instead of in Title Case.
**Author's response:** Changed the following reference titles to be in lower case rather than Title case. The authors also abbreviated the journal title in the reference Millero et al., 2008.

15   The references are now as follows:
Font, A., Prietsman, M., Tremper, A., Carslaw, D., and Green, D. C.: Identifying key sources of emissions of problem pollutants in Wales: a research collaboration to utilise novel monitoring and data analysis assessment techniques to drive innovation - Pontardawe Report, Environmental Research Group, King's College London, London, 2017.

20   Middlebrook, A. M., Bahreini, R., Jimenez, J. L., and Canagaratna, M. R.: Evaluation of composition-dependent collection efficiencies for the aerodyne aerosol mass spectrometer using field data, Aerosol Sci. Tech., 46, 258-271, 2012.
Millero, F. J., R. Feistel, D. G. Wright and McDougall, T.J.: The composition of standard seawater and the definition of the reference-composition salinity scale." Deep Sea Res. Part I: Oceanogr. Res. Pap. 55(1): 50-72,

25   2008.
Ng, N. L., Herndon, S. C., Trimborn, A., Canagaratna, M. R., Croteau, P. L., Onasch, T. B., Sueper, D., Worsnop, D. R., Zhang, Q., Sun, Y. L., and Jayne, J. T.: An aerosol chemical speciation monitor (ACSM) for routine monitoring of the composition and mass concentrations of ambient aerosol, Aerosol Sci. Tech., 45, 780-794, 2011.

30   US-EPA: Determination of metals in ambient particulate matter using X-Ray Fluorescence (XRF) Spectroscopy. Agency, U. S. E. P. A. (Ed.), U.S. Environmental Protection Agency, Cincinnati, OH 45268, 1999.

Page 14, lines 19-21: this reference should be updated; the article was already published in AMT.
**Author's response:** The reference was updated and is now listed as:

35   "Furger, M., Minguillón, M. C., Yadav, V., Slowik, J. G., Hüglin, C., Fröhlich, R., Petterson, K., Baltensperger, U., and Prévôt, A. S. H.: Elemental composition of ambient aerosols measured with high temporal resolution using an online XRF spectrometer, Atmos. Meas. Tech., 10, 2061-2076, 2017."

Page 17, caption of Figure 3: the measurement location should be added in the caption.

40   **Author's response:** Included the sample location to the caption of Figure 3. The caption now reads:
"Figure 3: Timeseries of K (top) and Ba (bottom) concentration ($\mu g\ m^{-3}$) using hourly XACT and daily ICP-MS measurements at Marylebone Road, London"

Page 18, caption of Figure 5: the measurement location should be added in the caption.

45   **Author's response:** Included the sample location and year to the caption of Figure 5. The caption now reads:
"Figure 5: Timeseries of non-sea salt $SO_4$ concentration (XACT, calculated) and non-refractory $SO_4$ (ACSM, measured) in ng m$^{-3}$ at Marylebone Road, London"

Page 19, caption of Figure 6: the measurement location should be added in the caption.

50   **Author's response:** Included the sample location and year to the caption of Figure 6. The caption now reads:
"Figure 6: Deming regression of water soluble Ca (top left), Cl (top right), K (bottom left) and $SO_4$ (bottom right) as measured by URG and Ca, Cl, K and calculated $SO_4$ (from elemental S) measured by XACT (ng m$^{-3}$) at Marylebone Road, London"

55

Supplement:
Page 2, top line: replace "sample location" by "sampling locations"
**Author's response:** Changed the title of supplement S1 as requested. It now reads:
"S1.      Map of the United Kingdom with sampling locations"

Page 2, caption of Figure S1: replace "sample locations" by "sampling locations".
**Author's response:** Changed the figure caption as requested. It now reads:
"Figure S1: Sampling locations in the field experiments"

**Author's response:** Some additional changes have been made to the manuscript after noticing punctuation errors:
P11L7 – Double full stop deleted in the sentence ending "…differences observed in concentrations."
10 P14L31 – Comma added between the report title and institution:

[revised manuscript text omitted]

All authors grant Copernicus Publications an irrevocable non-exclusive licence to publish the article electronically and in print format and to identify itself as the original publisher. Further, the authors grant Copernicus Publications commercial rights to produce hardcopy volumes of the journal for sale to libraries and individuals. The authors also grant any third party the right to use the article freely as long as its original authors and citation details are identified.

**10    Data Availability**

ICP-MS measurements on filters made at Pontardawe and Sheffield are available from https://uk-air.defra.gov.uk/data/metals-data. Additional datasets are available upon request to the corresponding author.

**11    Supplement link**

The Supplement related to this article is available online at <enter link when available>

**12    Author Contribution**

Anja H. Tremper - experiment design (XACT filter analysis), field and laboratory experiments, data ratification and analysis, manuscript preparation with contributions from all co-authors

Anna Font - field experiments and data analysis

Max Priestman - field experiments and ACSM data ratification

Samera H Hamad - field experiments (Marylebone Road)

Tsai-Chia Chung - laboratory experiments (calibration test of Cl, K and S)

Ari Pribadi - laboratory experiments (filter analysis)

Richard J. C. Brown - uncertainty calculations

Sharon L. Goddard - filter analysis (Wales/Sheffield)

Nathalie Grassineau - filter digestion (Marylebone Road)

Krag Petterson - technical input for XACT

Frank J. Kelly – manuscript review

David C. Green - experiment design (laboratory calibration test), field and laboratory experiments, uncertainty calculations, manuscript overview

**13    Competing interests**

Krag Petterson is employed by Cooper Environmental Services, the manufacturer of the instrument, and had input into the manuscript preparation from a technical perspective.

**14   Disclaimer**

This manuscript has not been published and is not under consideration for publication elsewhere.

---

## Author Response (AR3)

**Associate Editor Decision: Publish subject to minor revisions (review by editor)**
(28 Apr 2018) by Willy Maenhaut

Comments to the Author:
5  The following alterations are still needed for the main text before the manuscript can be published in AMT:

**Author's response:** The authors would like to thank the editor for the response and for carefully looking through the manuscript again. Please find below the detailed response to the comments.
10
Page 9, line 5: rename "Table 3-6" by "Tables 3-6".
**Author's response:** This was corrected in the last iteration but as a hyperlink was used, it possibly converted back to Table 3-6. The hyperlink was removed and it now reads:
"An overview of the data recorded in each comparison is given in Tables 3-6 and includes the limit of detection
15  for all elements."

Page 10, line 19: rename "Table 3-5" by "Tables 3-5".
**Author's response:** This was corrected in the last iteration but a hyperlink was used, it possibly converted back to Table 3-5. The hyperlink was removed and it now reads:
20  "LODs were not consistently higher for either the ICP-MS or the XACT measurements (Tables 3-5)."

Page 14, line 32: the title of the journal article should be in lower case.
Page 14, line 32: the reference "Hamad et al., 2016" should start on a new line.
**Author's response:** Changed the reference title to be in lower case rather than Title case and moved reference
25  to start in a new line:

[revised manuscript text omitted]

All authors grant Copernicus Publications an irrevocable non-exclusive licence to publish the article electronically and in print format and to identify itself as the original publisher. Further, the authors grant Copernicus Publications commercial rights to produce hardcopy volumes of the journal for sale to libraries and individuals. The authors also grant any third party the right to use the article freely as long as its original authors and citation details are identified.

**10 Data Availability**

ICP-MS measurements on filters made at Pontardawe and Sheffield are available from https://uk-air.defra.gov.uk/data/metals-data. Additional datasets are available upon request to the corresponding author.

**11 Supplement link**

The Supplement related to this article is available online at <enter link when available>

**12 Author Contribution**

Anja H. Tremper - experiment design (XACT filter analysis), field and laboratory experiments, data ratification and analysis, manuscript preparation with contributions from all co-authors

Anna Font - field experiments and data analysis

Max Priestman - field experiments and ACSM data ratification

Samera H Hamad - field experiments (Marylebone Road)

Tsai-Chia Chung - laboratory experiments (calibration test of Cl, K and S)

Ari Pribadi - laboratory experiments (filter analysis)

Richard J. C. Brown - uncertainty calculations

Sharon L. Goddard - filter analysis (Wales/Sheffield)

Nathalie Grassineau - filter digestion (Marylebone Road)

Krag Petterson - technical input for XACT

Frank J. Kelly – manuscript review

David C. Green - experiment design (laboratory calibration test), field and laboratory experiments, uncertainty calculations, manuscript overview

**13 Competing interests**

Krag Petterson is employed by Cooper Environmental Services, the manufacturer of the instrument, and had input into the manuscript preparation from a technical perspective.

**14  Disclaimer**

This manuscript has not been published and is not under consideration for publication elsewhere.